# Inner core static tilt inferred from intradecadal oscillation in the Earth's rotation

Yachong An [1,3], Hao Ding [1,3] ✉, Zhifeng Chen[1], Wenbin Shen[1] & Weiping Jiang [2] ✉

The presence of a static tilt between the inner core and mantle is an ongoing discussion encompassing the geodynamic state of the inner core. Here, we confirm an approximate 8.5 yr signal in polar motion is the inner core wobble (ICW), and find that the ICW is also contained in the length-of-day variations of the Earth's rotation. Based on the determined amplitudes of the ICW and its good phase consistency in both polar motion and the length-of-day variations, we infer that there must be a static tilt angle $\theta$ between the inner core and the mantle of about $0.17 \pm 0.03°$, most likely towards ~90°W relative to the mantle, which is two orders of magnitude lower than the 10° assumed in certain geodynamic research. This tilt is consistent with the assumption that the average density in the northwestern hemisphere of the inner core should be greater than that in the other regions. Further, the observed ICW period ($8.5 \pm 0.2$ yr) suggests a $0.52 \pm 0.05$ g/cm$^3$ density jump at the inner core boundary.

The oblate solid Earth consists mainly of a solid inner core, a liquid outer core, and a solid mantle with the same center of mass, which is reduced to a series of layered elliptical surfaces on which the density is constant in the classical model[1,2]. The current theories regarding the Earth's rotation involve the consideration of the mantle's elliptical surfaces of constant density, whose symmetry axes are aligned in the direction of rotation, and hydrostatic effects necessitate that the inner core's figure axis $\Omega_{ic}$ (as defined in Appendix A of ref. 2) and rotation axis $\Omega'_m$ are aligned with the mantle's figure or rotation axis $\Omega_m$ (which are nearly identical due to centrifugal torque) in order to maintain equilibrium. The presence of random torques acting on the inner core results in a slight tilt and further excites a prograde rotation mode known as the inner core wobble (ICW), i.e., the inner core's figure axis $\Omega_{ic}$ wobbles about its rotation axis $\Omega'_m$[2-4] (also represents the direction of the lowest gravitational potential energy of the mantle-inner core system). The above 'tilt' between the $\Omega_{ic}$ and $\Omega'_m$ is a generally dynamic tilt, and in this case, the ICW theoretically appears only in the polar motion (PM) of the Earth's rotation[2,3,5]. In addition, the ICW period is very sensitive to the density jump $\Delta\rho_{ICB}$ at the inner core boundary

(ICB; $\Delta\rho_{ICB}$ is still not well constrained[6]), and based on the PREM model[1], theory predicts that its period falls in the range of 6.6–7.8 yr[3,7–9]. However, the elliptical surfaces of constant density within the heterogeneous mantle may exhibit random tilting around its rotation axis $\Omega_m$ considering its solid properties, particularly with significant uncertainties at the core-mantle boundary (CMB). Consequently, the inner core's rotation axis $\Omega'_m$, which signifies the direction of the static equilibrium of the inner core (and corresponds to the lowest gravitational potential energy of the mantle-inner core system[10]), was previously believed to deviate from alignment with the mantle's axis $\Omega_m$ and instead possess a tilt relative to it, and thereby the 'tilt' between the $\Omega'_m$ and $\Omega_m$ is called as static tilt. To explain the decadal oscillations in both the PM and the length-of-day variations ($\Delta$LOD) as the possible ICW, the inner core's rotation axis $\Omega'_m$ was proposed to coincide with the dipole axis of the geomagnetic field (tilted 10° westwards from $\Omega_m$)[11]. Despite the absence of confirmed observations of the ICW[12,13] and the lack of universal acceptance of the excessive static tilt of 10°, the possibility of a static-tilted inner core remains, and further investigation has been conducted to explore the

[1]School of Geodesy and Geomatics, Hubei LuoJia Laboratory, Wuhan University, 430079 Wuhan, China. [2]GNSS Research Center, Wuhan University, 430079 Wuhan, China. [3]These authors contributed equally: Yachong An, Hao Ding. ✉e-mail: dhaosgg@sgg.whu.edu.cn; wpjiang@whu.edu.cn

impact of a static tilt on the period of the ICW[12]. Theoretically, such a static tilt must affect some modes that are sensitive to the inner core. However, no relevant eigenfrequency deviation has been clearly detected in the core-sensitive normal modes of Earth's free oscillation[14,15], which denotes that this static tilt is still uncertain, it may not exist or is quite small. Overall, a statically tilted inner core will be of great importance to some fundamental research about the Earth, such as the differential rotation of the inner core, the Earth's surface gravity changes, the seismic tomography of the deep Earth, and the Geodynamo theory[6,12,14,16–20].

A statically tilted inner core will induce changes in the rotational normal modes of the Earth, with the ICW being the most sensitive. In the presence of a statically tilted inner core, the ICW will manifest not only in the PM but also in the ΔLOD[11]. By identifying a similar periodic signal in the ΔLOD and establishing its correlation with the ICW identified in the PM, we can, in turn, ascertain the presence of a statically tilted inner core. Furthermore, the angle $\theta$ of static tilt can be determined by comparing the corresponding amplitudes of the two signals (in the PM and in the ΔLOD, respectively).

## Results and discussion

### Inner core wobble in the polar motion and the length-of-day variations

In this study, we report the results from the ΔLOD and PM time series. The chosen ΔLOD time series is a yearly time series with a 1900–2020 time span. For the PM time series, the 1900–2020 EOPC01 time series with one-year sampling is used ($x$ and $y$ components). The pretreatments of the ΔLOD and PM time series are shown in the Methods. Figure 1 shows the ΔLOD and PM records used. For the periodic signals present in the PM and ΔLOD, the consensus is that they are excited by the Earth's internal or external sources through the conversion of angular momentum[21]. Hence, we need to rule out the influence of external excitation sources before determining that a target signal is from the Earth's internal motion. There are three external excitation sources of the PM and ΔLOD changes, the atmospheric, oceanic, and hydrological effects. Of these, the first two effects are the two main external excitation sources[21–23]; although hydrological effects will also excite the Earth's rotation changes, previous studies have proven that the hydrological effects have no significant contribution to the target 5.5–10 yr period band[13,23] and different hydrological models have clear deviations[13,24]. Hence, similar to previous studies[13,22], we only consider

the atmospheric and oceanic effects. The PM and ΔLOD excited by the atmospheric angular momentum (AAM) and oceanic angular momentum (OAM) are also shown in Fig. 1. Supplementary Fig. 1 (in the Supplementary Information) also shows all datasets used before combining.

Different from the oceanic tidal signals that have both prograde and retrograde components in PM[25–27], the ICW is a prograde motion (the same as the Chandler wobble, i.e., the mantle wobble); in the complex spectra of the PM, a prograde/retrograde wobble only has a positive/negative frequency. The identification of the well-known Chandler wobble is based on this feature[28–30]. Therefore, as a prograde motion, the ICW only appears on the positive frequency axis of the PM spectrum and this is a distinguishing feature for identifying it. Based on the 1960-2017 PM record without removing the AO (AAM + OAM) effects, a previous study[13] used this feature to identify an ~8.7 yr signal for the ICW; here, we perform independent detection by using a longer record (1949–2020) and further consider the AO effects. Figure 2 shows the normalized AR-z spectra (see Methods) of the PM and ΔLOD records in the 1949–2020 time span, in which the AO effects have been removed. Figure 2a shows four different harmonic signals (-5.9, -7.3, -8.5, and -9–11 yr) in the positive frequency axis; only the ~8.5 yr signal has no corresponding spectral peak in the negative frequency axis (The AR-z method is meant for determining the presence of a signal and estimating its frequency, the amplitude of it contains no direct information about the actual complex amplitude of the detected signal). The corresponding Fourier spectra of the PMs (observed and AO excited) show similar findings (see Supplementary Fig. 2b). Among these harmonics, the ~5.9 yr signal has been suggested as the inner core oscillation coupled with torsional wave in the Earth's core but still remains controversial[31–33]; the ~7.3 yr signal can be interpreted as the Magneto-Coriolis eigenmode in the Earth's core based on a theoretical model[34]; the peak in the ~9-11 yr is possible from the ~11 yr Schwabe solar cycle. The adjacent ~13 yr signal (Fig. 2a) also has both positive and negative frequencies; similar period was also found in the geomagnetic dipole field[35], but the underlying mechanism is still enigmatic; the ~18–23 yr spectral peak may be mainly caused by the 18.6 yr tidal signal and the ~22 yr Hale solar cycle or high-latitude MAC

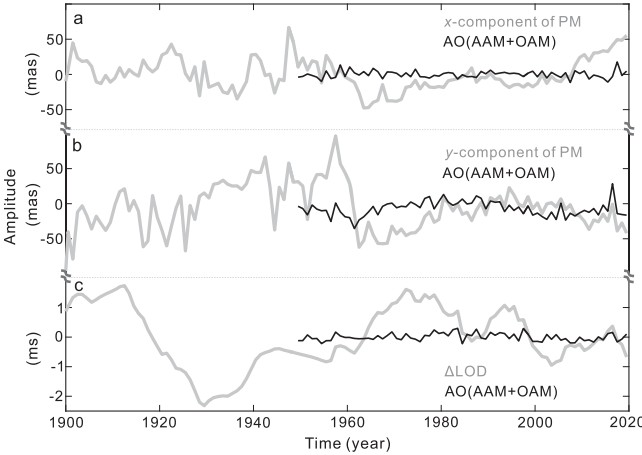

**Fig. 1 | The used polar motion (PM) and length-of-day variations (ΔLOD) records.** The $x$ component (**a**) and $y$ component (**b**) of the observed PM records from 1900 to 2020 (gray curves) and the PM excited by the atmospheric and oceanic angular momentum (AAM + OAM, abbreviated as AO: black curve) from 1949 to 2020; **c** the observed ΔLOD record from 1900 to 2020 (gray curves) and the ΔLOD excited by the AAM + OAM (AO: black curve) from 1949 to 2020.

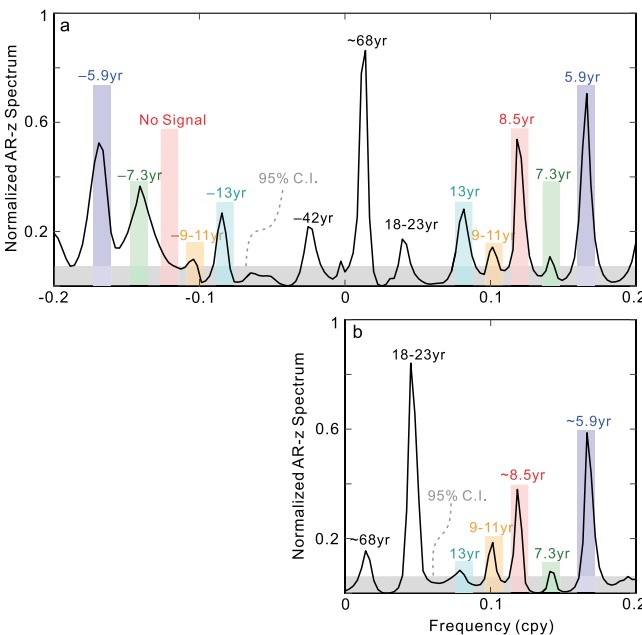

**Fig. 2 | Normalized AR-z spectra of the used polar motion (PM) and length-of-day variations (ΔLOD) records (1949–2020).** The AR-z spectra of the PM (**a**) and ΔLOD (**b**) records in which the atmospheric and oceanic angular momentum effects were removed. Different colored rectangles indicate different periodic signals; the gray area represents 95% confidence intervals (C.I.).

(magnetic-Archimedes-Coriolis forces) wave[36] in the Earth's core; those periods are too long to be the ICW[3,7–9]. Therefore, the 8.52 ± 0.19 yr signal is the only candidate for the ICW. Since no other mechanism has been proposed to account for such a prograde ~8.5 yr motion, and the AO effects have been removed, we can conclude that the 8.5 yr signal is the ICW. In addition, the uncertainties for the estimates in this study were based on a bootstrap procedure[37].

Comparing Fig. 2a and b, a finding is that the six periodic/quasi-periodic signals in the positive frequency axis of the PM spectra are also present in the spectrum of the ΔLOD. This mainly benefits from the high-frequency resolution of the AR-z spectrum and its strong sensitivity to harmonic signals[38]; the Fourier spectra can only identify parts of those signals (see Supplementary Fig. 2). These consistencies deserve further attention, but we only focus on the 8.5 yr signal (the ICW signal). Figure 2b confirms that the ICW signal is also present in the ΔLOD (with an 8.47 ± 0.32 yr period); this finding preliminarily suggests that there should be a static tilt between the inner core and the mantle. Given that the AO effects have no significant contribution to the target signal, we use the 1900–2020 PM and ΔLOD records to extract the ~8.5 yr signal to obtain higher resolutions. For simplicity, we directly use a cosine least-square fitting process.

### Static tilt between the inner core and mantle

To further obtain the orientation of the static tilt angle $\theta$ and its magnitude, we need to determine the fluctuation characteristics of the axial torque $\Gamma_z$ ($\propto d\Delta\text{LOD}/dt$; see Methods) exerted on the mantle. Hence, we directly fit the ~8.5 yr signal from $d\Delta\text{LOD}/dt$; the fitted results from ΔLOD can be found in Fig. S3.

Figure 3 shows the fitted ICW from the $d\Delta\text{LOD}/dt$ and the $x$ and $y$ components of the PM. Clearly, the ICW from the $y$-component is ahead of that from the $x$-component by ~$\pi/2$ (see green areas in Fig. 3); since the directions of $x$ and $y$ have a $\pi/2$ angle difference in the equatorial plane, these findings are acceptable. The most important

point obtained from Fig. 3 is that, for the first time, we find that the ICW signals contained in the $y$ component of the PM and $d\Delta\text{LOD}/dt$ have almost synchronous phases; the extracted oscillations using a more complicated method (the normal time-frequency transform, NTFT[39]) show almost the same results (see Supplementary Fig. 3). This synchronicity is not a random phenomenon and at least demonstrates that the inner core tilts in a particular direction (see the possible scenario in Fig. 4). The axial torque $\Gamma_z$ reaches its peak/trough only when the $\Gamma_z$ is in the plane defined by the static tilted axis $\Omega'_m$ and the rotation axis of the mantle $\Omega_m$; hence, we can deduce that the inner core tilts should be along the ~90°E–90°W direction.

Given the $y$ component of the PM along the 90°W longitude, the phase synchronization in Fig. 3a, c indicates that the inner core is more

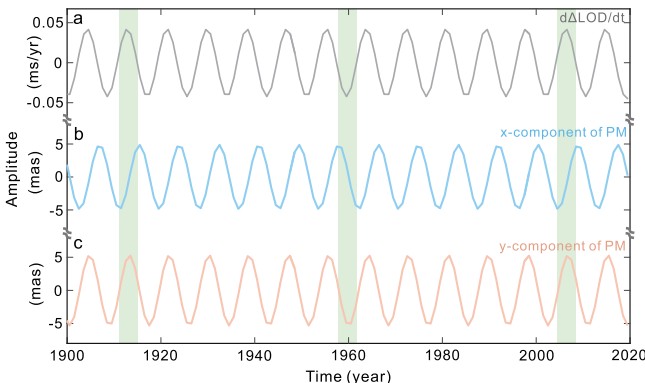

**Fig. 3 | The least-square fitted ~8.5 yr signals from different records.** From (**a**) the first-order time derivative of length-of-day variations ($d\Delta\text{LOD}/dt$); (**b**) the $x$ component of the polar motion (PM), and (**c**) the $y$ component of the PM. The time span used is 1900–2020, and the sampling interval is one year.

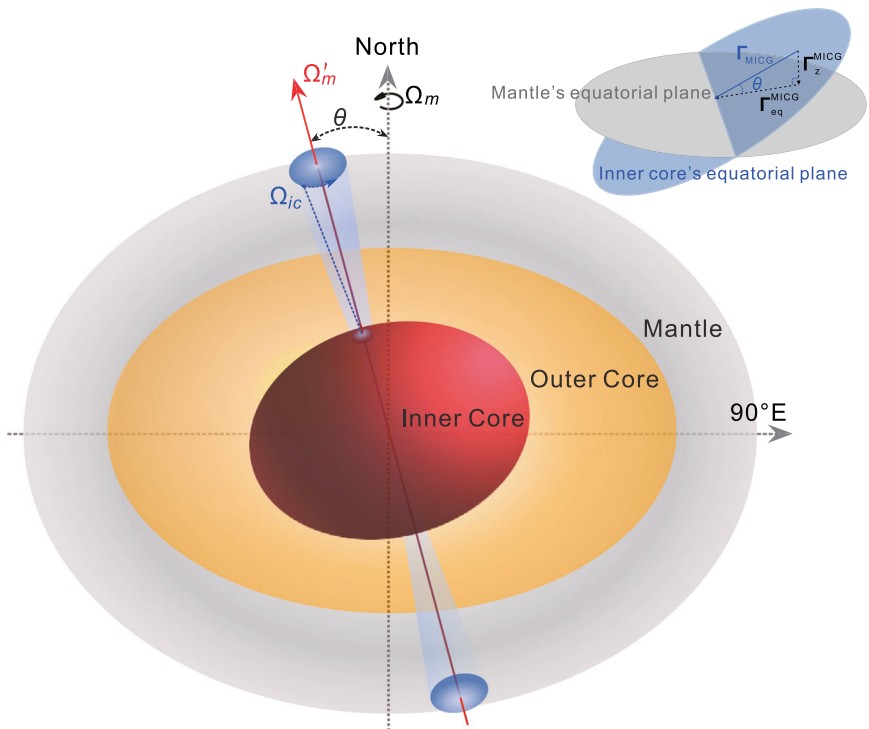

**Fig. 4 | Schematic depiction of the tilted inner core wobble (ICW).** The figure axis of the inner core wobbles about its rotation axis tilted to the mantle in a small circular orbit (see blue shaded area). $\Omega_m$ is the rotation axis of the mantle, $\Omega'_m$ and $\Omega_{ic}$ are respectively the rotation axis and dynamic figure axis of the inner core, and $\theta$ is the static tilt angle between $\Omega_m$ and $\Omega'_m$. The upper right subgraph shows how to get $\theta$ from the mantle-inner core gravitational torque in the mantle's axial and equatorial directions $|\Gamma_z^{MICG}|$ and $|\Gamma_{eq}^{MICG}|$ (see also in Methods).

likely tilted in the 90°W direction, which is also similar to that suggested by previous studies[11,12,40,41]. In terms of the long-term dynamic conservation of the Earth's angular momentum, this static westwards tilt is consistent with the effect of the non-axisymmetric mass of the inner core, i.e., the western hemisphere (more specifically, the northwestern hemisphere) of the inner core should have greater average densities. To explain the asymmetry between the inner core's eastern and western hemispheres in seismological observations[42,43], a previous dynamical model considers the crystallization and melting at the surface of the inner core and has similar suggestions[44], i.e., the western hemisphere of the inner core is denser than its eastern hemisphere. Interestingly, a seismological study has suggested that the western hemisphere of the inner core may be relatively denser[45], and a thicker compacting layer at the top of the inner core's western hemisphere was also suggested[46]; a more nuanced research found that the western zone is largely confined to the northern hemisphere[47]; those suggestions are generally consistent with the westwards statically tilted inner core that we found.

Here we can propose the following scenario as schematically depicted in Fig. 4: There is a static westward tilt $\theta$ between the inner core and the mantle, resulting in the inner core exhibiting a wobbling motion around the tilted axis $\Omega'_m$. This wobbling motion leads to the exchange of angular momentum between the mantle and inner core in both the equatorial and axial directions of the mantle, consequently giving rise to the appearance of the ICW in both the PM and ΔLOD. The inner core and mantle then obtain the maximum or minimum deviation at ~90°E − 90°W (i.e., $x \approx 0$) equatorial diameter; the torque on the inner core (equatorial plane of the inner core) thus has a maximum/minimum component on the axis of rotation when $x \approx 0$. At the same time, the ΔLOD is also 0 due to a first derivative relationship with the exchange of the axial angular momentum (see Supplementary Fig. 3). Therefore, there will be a good phase consistency between the $d\Delta\text{LOD}/dt$ and the $y$ component of the PM for the ICW (confirmed in Fig. 3).

Next, we estimate the tilt $\theta$. The heterogeneous mantle results in a tilted rotation axis of the inner core $\Omega'_m$. When the inner core's figure axis $\Omega_{ic}$ is aligned with $\Omega'_m$, the mantle-inner core system has the lowest mantle-inner core gravitational (MICG) potential energy (the equilibrium state). Any deviation of the inner core's figure axis $\Omega_{ic}$ from the equilibrium state caused by the ICW will result in a MICG restoring torque $\boldsymbol{\Gamma}_{\text{MICG}}$; the torque $\boldsymbol{\Gamma}_{\text{MICG}}$ always brings the inner core back to the equilibrium state. Thus, the torque $\boldsymbol{\Gamma}_{\text{MICG}}$ is always in the plane (the blue plane in Fig. 4) perpendicular to the inner core's rotation axis $\Omega'_m$. The mean equatorial plane of the inner core is perpendicular to the axis $\Omega'_m$, so $\boldsymbol{\Gamma}_{\text{MICG}}$ is always in the mean equatorial plane of the inner core. This MICG torque can also be decomposed into the equatorial and axial torques exerted on the mantle due to the tilt of the inner core. If there is no static tilt between the inner core and mantle, the axial torque exerted on the mantle will be 0, and the entire gravitational torque will provide the equatorial torque exerted on the mantle.

Under the MICG coupling, the equatorial torque exerted on the mantle can be written as the following (see Methods):

$$\left|\boldsymbol{\Gamma}_{\text{eq}}^{\text{MICG}}\right| = (C_m - A_m)\Omega_0^2 |\boldsymbol{\chi}_{\text{ICW}}(t)| \tag{1}$$

where $\boldsymbol{\chi}_{\text{ICW}}(t)$ is the excitation sequence of the ICW in the PM, and the observed ICW is almost the same as its excitation sequence due to the low frequency[21,48] (see Methods). Since the ICW identified in the PM has an amplitude of $4.7 \pm 0.4$ mas, we can calculate $|\boldsymbol{\Gamma}_{\text{eq}}^{\text{MICG}}|$ to be $(2.87 \pm 0.24) \times 10^{19}$ N·m. According to the theorem of angular momentum, the axial torque exerted on the mantle is written as the following (see Methods):

$$\left|\boldsymbol{\Gamma}_z^{\text{MICG}}\right| = \frac{\Omega_0 C_m}{\text{LOD}} \frac{d(\Delta\text{LOD}_{\text{ICW}})}{dt} \tag{2}$$

Substituting the amplitude of $0.046 \pm 0.005$ ms/yr (corresponding to the amplitude of $0.061 \pm 0.007$ ms of $\Delta\text{LOD}_{\text{ICW}}$) and the period of 8.5 yr for the observed ICW signal in the ΔLOD into Eq. (2), the axial torque $|\boldsymbol{\Gamma}_z^{\text{MICG}}|$ is calculated to be $(8.61 \pm 0.95) \times 10^{16}$ N·m; this is only a small component of the equatorial torque of the inner core caused by the ICW due the inner core static tilt. Therefore, the static tilt angle $\theta$, or the angle between the axis about which the inner core wobbles and the rotation axis of the mantle, is calculated as $\arctan(|\boldsymbol{\Gamma}_z^{\text{MICG}}|/|\boldsymbol{\Gamma}_{\text{eq}}^{\text{MICG}}|)$ (see Fig. 4) and equal to $0.17 \pm 0.03°$; this is much smaller than previous assumptions.

Our observed ICW period is slightly larger than the theoretical values (6.6–7.8 yr)[3,7–9], but considering that even the generally accepted Chandler wobble observation period of prograde ~430 days is ~30 days longer than its theoretical periods[28–30], free core nutation observation period of retrograde ~430 days is ~20 days shorter than its theoretical periods[49,50], and that the density jump $\Delta\rho_{\text{ICB}}$ at the ICB was also poorly determined[6,51], this deviation is accepted. Considering this newly determined period of the ICW, we can also invert the density jump $\Delta\rho_{\text{ICB}}$. Taking the density profiles of the PREM model as a reference, we finally obtained $\Delta\rho_{\text{ICB}} = 0.52 \pm 0.05$ g/cm³ (see Methods), which is smaller than that of the PREM model (0.598 g/cm³).

In summary, based on the Earth's rotation observations (PM and ΔLOD), we experimentally confirmed for the first time that the 8.5 yr signal is the ICW. The evidence indicates that the inner core is tilted to the mantle along ~90°W, and the inverted tilt angle is $0.17 \pm 0.03°$; this static tilt angle means that the average density in the northwest hemisphere of the inner core should be greater. The larger observed period may also indicate that the eastwards differential rotation rate of the inner core should be much less than 1° per year[12,16]. Besides, the density jump of $0.52 \pm 0.05$ g/cm³ at the ICB is also inverted based on the observed ICW period. Undeniably, it is difficult for seismological observations to detect such inner core static tilt directly, but interestingly, the results from seismological studies showed that the western/northwestern hemisphere (or at least its top layer) of the inner core may be relatively denser[42–47]. These suggestions, although they have some uncertainties, are qualitatively consistent with our finding of a westwards-tilted inner core, and we suggest such consistency should be helpful to the inner core oscillation or differential rotation.

## Methods
### Conservation of angular momentum of the mantle and inner core
Considering the mantle alone, the law of angular momentum can be rewritten as[48]:

$$\frac{d}{dt}\mathbf{H}_m + \boldsymbol{\Omega} \times \mathbf{H}_m = \boldsymbol{\Gamma}_m \tag{3}$$

where Earth's angular velocity $\boldsymbol{\Omega} = \Omega_0[m_1, m_2, 1+m_3]^{\text{T}}$; the mantle angular momentum is:

$$\mathbf{H}_m = \mathbf{I}_m \boldsymbol{\Omega} \tag{4}$$

The asymmetric part of the mantle mentioned above is insignificant relative to its axisymmetric part, and the mantle can still be approximately as axisymmetric in the calculation of torque for simplicity; $\mathbf{I}_m$ is the mantle moment of inertia tensor initially expressed in the principal axes:

$$\mathbf{I}_m = \begin{bmatrix} A_m & 0 & 0 \\ 0 & A_m & 0 \\ 0 & 0 & C_m \end{bmatrix} \tag{5}$$

Combining the above equations with the eigenfrequency of the free Euler wobble replaced by the Chandler wobble $\sigma_{\text{CW}}$, the equatorial

torque exerted on the mantle can be obtained by the observed PM

$$\frac{i}{\sigma_{CW}}\frac{d\mathbf{m}}{dt} + \mathbf{m} = \boldsymbol{\chi}(t) = \frac{\boldsymbol{\Gamma}_{eq}}{i(C_m - A_m)\Omega_0{}^2} \qquad (6)$$

where $\boldsymbol{\Gamma}_{eq} = \Gamma_1 + i\Gamma_2$ is the equatorial torque exerted on the mantle; $\Omega_0 = 7.29212 \times 10^{-5}$ s$^{-1}$ is the mean (sidereal) rotation rate[9]; $A_m = 7.0999 \times 10^{37}$ kg·m$^2$ and $C_m = 7.1236 \times 10^{37}$ kg·m$^2$ are the equatorial and axial moments of inertia of the mantle, respectively; $\mathbf{m} = m_1 + im_2 = x -iy$ is the observed PM; $\boldsymbol{\chi}(t) = \chi_1 + i\chi_2$ is its excitation function. The relation between an excitation function of complex frequency $\sigma$ and the motion of the observed pole is $\boldsymbol{\chi}(t) = (1-\sigma/\sigma_{CW})\mathbf{m}$ (in which $\sigma_{CW} = \omega_{CW} + i\gamma_{CW}$, $\omega_{CW} = 2\pi \cdot 0.843$ cpy and $\gamma_{CW} = \omega_{CW}/2Q_{CW}$; $Q_{CW} \approx 30\text{–}150$[52,53]. Thus, there is little difference between excitation and observation in the low-frequency band.

Similarly, the axial torque exerted on the mantle can be obtained by the following:

$$\Omega_0 C_m \frac{dm_3}{dt} = \boldsymbol{\Gamma}_z \qquad (7)$$

where $m_3 = -\Delta\text{LOD}/\text{LOD}$; LOD = 86400 s. Combining with Eqs. (1), (2), (6) and (7), we can directly infer the static tilted angle of the inner core from the ICW signal in the $\Delta$LOD and PM, which is impossible in related previous studies[11,12].

## Stabilized AR-z spectrum

A real discrete time series with the length of $N$ equally spaced samples, which contains $M$ harmonics, is written as (which satisfies the AR relation[54]):

$$x(n) = \sum_{j=1}^{M}\left[\mathbf{A}_j\exp(in\sigma_j) + \mathbf{A}_j^*\exp(-in\sigma_j^*)\right], n=1,2,3,\ldots,N \qquad (8)$$

where $\mathbf{A}_j = A_j\exp(i\phi_j)/2$ is the complex amplitude ($A_j$ and $\phi_j$ are the amplitude and initial phase) and $\sigma_j = \omega_j + i\alpha_j$ is the complex frequency of a given harmonic ($\omega_j$ and $\alpha_j$ are the angular frequency and decay rate). By using a frequency-domain AR method, the complex frequency $\sigma_j$ can be estimated[38]. A Lorentzien power spectrum in the complex $z$ plane can be formed as follows[38]:

$$P(\sigma_i) = \frac{1}{|\exp(\tilde{\sigma}_i) - \exp(i\sigma_i)|^2}, i=1,2,\ldots,N \qquad (9)$$

where $\tilde{\sigma}_i$ and $\sigma_i$ are the estimated and referred complex frequencies, respectively. For the specific execution of the stabilized AR-z spectrum, please see the Supplementary Text (in the Supplementary Information).

## Constraint for the density jump at the inner core boundary

The frequency of the ICW can be written as follows (in cpsd: cycle per solar day)[3]:

$$\sigma_{ICW} = [\alpha_3(1 + \alpha_g)(e_s + S_{34}^g + S_{34}^p)]/(1 + K^{ICB}) \qquad (10)$$

where the elastic compliances $S_{34}^g = -1.812 \times 10^{-6}$, $S_{34}^p = -2.686 \times 10^{-4}$ and $K^{ICB}$ is a dimensionless coupling constant[55] and Real($K^{ICB}$) = $1.11 \times 10^{-3}$; $\alpha_3$ and $\alpha_g$ have the following forms:

$$\begin{cases} \alpha_3 = 1 - (A'e'/A_s e_s)\alpha_g \\ \alpha_g = \frac{3G}{a_s^2\Omega^2}\left(\left[\frac{5\bar{\rho}}{3\rho_f}+1\right]A'e' - A_s e_s\right) - 1 \end{cases} \qquad (11)$$

where $\rho_f$ is the fluid density just outside the ICB, $\bar{\rho}$ is the mean density of the inner core, $A_s$ and $e_s$ are the equatorial moment of inertia and the

dynamical ellipticity of the inner core, respectively, and $A'$ and $e'$ have similar definitions but for a body of the inner core radius with the constant mass of that of the fluid core at the ICB[2].

$$\begin{cases} A_s = \frac{8\pi}{3}\int_0^{a_s}\rho(r)\{r^4 - \frac{1}{15}\frac{d[\varepsilon(r)\cdot r^5]}{dr}\}dr \\ A'e' = \frac{8\pi}{15}\rho_f a_s^5 \varepsilon_s \\ A'(1 + \frac{e'}{3}) = \frac{8\pi}{15}\rho_f a_s^5 \end{cases} \qquad (12)$$

in which $\varepsilon(r)$ is the geometrical ellipticity of the Earth and $a_s$ is the inner core radius. Taking the PREM model as a reference because it is the generally accepted model, underlying the conservation of the whole Earth's mass and angular momentum, we can modify the density of the outer core (based on the related expression given in PREM) and hence change the inner core density profile to obtain the observed ICW period. When the observed 8.5 yr period is obtained, the corresponding $\Delta\rho_{ICB}$ is the one we recommend using.

## Datasets and preprocessing

The PM observations were obtained from the EOPC01 dataset (1861/01-1889/12 with 0.1 yr sampling and 1900/01–2019/12 with 0.05 yr sampling); the $\Delta$LOD record was combined with a long-term dataset[56] (1623/06-2008/06 with 1 yr sampling from IERS; EOPC01) and the EOPC04 $\Delta$LOD record[57] (1962/01–2019/12 with 1-day sampling); the AAM (1949/01–2019/12, sampling at 6 h) record was from the Special Bureau for the Atmosphere[58–60]. The AAM was calculated from NCEP/NCAR reanalyses archived on pressure surfaces, and the inverted barometer (IB) pressure term was chosen as the mass term. The OAM record was obtained from the Special Bureau for the Oceans' datasets: ECCO_50 yr[61] (1949/01–2003/01, sampling at 10 days) and ECCO_kf080i[62] (1993/01–2020/3, sampling at 1 day). Those datasets are shown in Supplementary Fig. 1. To standardize the sampling intervals of the records, we down-sampled all records to 1 yr, and to avoid aliasing effects in this down-sampling process, a low-pass filter (with a cut-off frequency $f_c = 0.5$ cpy) was used prior to down-sampling.

Note that although the theoretical amplitudes of the 8.85 yr and 9.3 yr zonal tides are quite small (only ~2 μs for the 9.3 yr tide and <1 μs for the 8.85 yr tide) and far less than the background noise level of the $\Delta$LOD time series, they were removed from this $\Delta$LOD record based on a given model[63] to avoid the effects of some well-known signals on the target ~8 yr period band. The $d\Delta$LOD/$dt$ was obtained by a classical discrete numerical derivation algorithm, i.e., $d\Delta\text{LOD}(t_i)/dt = [\Delta\text{LOD}(t_{i+1}) - \Delta\text{LOD}(t_i)]/\Delta t$.

## Data availability

All associated source data including Earth's rotation and external excitations have been deposited in the Figshare database and can be accessed at https://doi.org/10.6084/m9.figshare.22820525; those data can be used to reproduce the results shown in Figs. 1–3.

## Code availability

The code of the AR-z spectrum has been uploaded to https://doi.org/10.1029/2018JB015890. It is also available from the corresponding author H.D upon request.

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

## Acknowledgements

The authors thank Wei Wang and B.F. Chao for their useful suggestions and discussion. H.D. was partially supported by the National Natural Science Foundation of China (NSFC) through awards 41974022, 42192531, and 41721003, the Special Fund of Hubei Luojia Laboratory through award 220100002. Y.A. was partially supported by the Key Laboratory of Geospace Environment and Geodesy, Ministry of Education, Wuhan University through award 210204.

## Author contributions

H.D. and W.J. supervised the project. Y.A. and H.D. conceived the idea and designed the experiments. Y.A. conceived the data analysis and wrote the first draft. H.D. prepared the figures, assisted in the overall conceptualization and interpretation of results, and contributed substantially to the writing of the manuscript. Z.C., W.J. and W.S. contributed to discussions of the data and analysis. All authors participated in the writing of the manuscript.

## Competing interests

The authors declare no competing interests.
