## [Peer Review File · Nature Communications]

REVIEWER COMMENTS

Reviewer #1 (Remarks to the Author):

A review of the paper:

Inner core static tilt: Evidence from intradecadal oscillation in the Earth's rotation
by An et al.

This is a paper on the possibility that the inner core has a static tilt with the Earth's rotation axis.

Consider the very first sentence of your manuscript.

“The Earth consists mainly of a solid inner core, a liquid outer core, and a solid inner core with the same center of mass.”

I don't know what the above sentence means, but the sloppiness, unfortunately, persists throughout the entire manuscript. Although geodynamics is not my area of expertise, and this paper needs to be assessed by someone with good knowledge of the current research on the tilt to judge the novelty you claim, you did not convince me that you can observe the tilt of the Earth's inner core, let alone prove it. There are two main reasons: firstly, in the abstract and elsewhere, you say that the tilt you infer is 0.17 degrees away from the Earth's spin axis. However, you do not present the uncertainty of that estimate, and given that it is probably quite large, it could be said that you obtained a 0.0-degree tilt or no tilt at all. Secondly, a correlation is not causation. You will need more robust evidence that the inner core has a tilt apart from showing the two spectra and deducing that the signal you observe in the spectrum is the only candidate for the ICW. Take, as an example, a recent claim by seismologists that the rotation of the inner core has some correlation with the LOD change – no one in the seismological and the broad geophysical community can take it as proof that the two are related or caused by the same phenomenon due to a significant uncertainty of those estimates and a lack of physical mechanism explaining the correlation. This lack of uncertainty analysis and solid proof that you are observing a tilt of the inner core makes your paper highly speculative. Furthermore, you use some seismological papers that showed complexities in the inner core to reinforce that the inner core might have a non-axisymmetric mass, but in my opinion, a more rigorous understanding of what these papers are all about is lacking from the manuscript.

So, a tilt and a differential rotation of the inner core might be real as they emerge intuitively and numerically from geodynamical modeling (as shown in multiple papers), but you need more in this paper to convince me that what you have achieved is a conceptual advance.

Let me comment further on your reference to the seismological papers.

You misunderstood the goal of most inner-core seismological papers you cite. The goal was not to establish a tilt of the inner core but to study anisotropy. Then, by making a strong assumption that anisotropy could be approximated by a cylindrical model, one can perform a grid search to see if the data one has would point to the main axis of anisotropy oriented in a slightly different direction than the spin axis of the Earth. You can then obtain some mean, which cannot be oriented exactly along the Earth's rotation axis because of the noise in the travelttime data. What's worse than that is that we don't have an even volumetric sampling of the ray paths sensitive to the inner core. You cite the paper by Su et al. (1996) and their estimate of the tilt within the range of 5–10 degrees, but you are perhaps unaware that this tilt was shown to be meaningless by Souriau et al. (1997) because of the bias in the data. By the way, they didn't find any wobble in time, even with that biased estimate. The bottom line is that these early estimates of the anisotropy axis were shown to be unreliable and erroneous. For that reason, they were abandoned, and the differential rotation of the inner core estimates were conducted by other methods. Therefore, your underlying thesis that your estimate of the tilt is two orders of magnitude lower than what is obtained by seismology is wrong and misleading. Seismology has not yet determined the tilt of the inner core or anything like that.

Also, let me comment on your invoking seismology as a method that found that the mass distribution in the Earth's inner core is not axisymmetric. While heterogeneities are documented to exist near the inner core boundary, everything beyond that is much more speculative. If we accept that one hemisphere of the inner core is faster than the other, would that necessarily indicate a denser material and more mass? Why would anisotropic fabric be a denser fabric? The opposite can be true. From seismology, anisotropy appears weaker in the outer part of the inner core and stronger towards its center. It does not necessarily have anything to do with cylindrical anisotropy's fast and slow axis but with the pronounced slow axis in the innermost part of the inner core. For instance, deeper into the inner core, two recent studies that you cited show an offset of the anisotropic (as hypothesized) volume from the center at two different locations of the inner core. In all honesty, there is no consensus as to which one is correct, so your inference that a tilt is to the western side needs more seismology support. But if you honestly present seismological work as it is, you will find it more difficult to substantiate your claims.

There is no such thing as "super-rotation". (Line 34) We talk about the differential rotation of the inner core relative to the mantle. Besides, the paper would benefit from a discussion how the inner core tilt would impact the differential rotation studies.

The quasi-eastern and quasi-western anisotropy is not that established. The robust differences are documented in absolute velocity between the eastern and the western hemisphere, but only at the top of the inner core. Given the sparse coverage of the inner core on the western side, whether the same holds for anisotropy is hard to tell. As you go deeper toward the center, the hemispherical division loses its meaning, and there is no consensus that any hemispherical pattern exists.

Reviewer #2 (Remarks to the Author):

First of all, I would like to state for the record that I believe that the authors have clearly and convincingly demonstrated that the results they present in the article are true. Moreover, I would like to add that in my view they are truly important and sound. Applying a methodology that has already shown its power in other applications to geodetic problems and using precise earth rotation data, they have proved that the axis of the earth's solid inner core is tilted with respect to the polar axis of the whole earth, and at the same time they have determined for the first time empirically the value of the so-called ICW period. This motion can be thought of as analogous to the precession motion of the earth under lunisolar attraction, but in this case it is the inner core under the attraction of the earth's mantle surrounding it. This free mode of the Earth's rotation has so far only been known from theoretical developments and the value of its period inferred from geophysical models, as well as more indirectly from its effects on the Earth's rotation, more precisely on nutation, since the introduction in 1991 of a basic earth model made of 3 ellipsoidal, concentric and aligned layers, which was later refined and fitted to VLBI data, becoming the official IAU2000 nutation model.

Although the authors only highlight the importance of the experimental determination of the inner core inclination from earth rotation data, more precisely from pole motion (PM) and length-of-day (LOD), I consider that the determination of the value of this ICW period is not less important than the previous finding and I could even dare to say that for me it could be even more important in that the accuracy of the method they use to identify weak harmonic signals (stabilized AR-z spectrum) is very precise in determining the value of the periods, but does not allow determining the amplitudes with the same relative accuracy as far as I know.

In my opinion, there is no doubt about the veracity of the results, since they clearly identify a harmonic constituent of the pole motion with a period close to 8.5 years, formed only by the prograde component, and also an oscillation of smaller amplitude in LOD with the same period; current theoretical knowledge can only explain this coincidence if the origin of the signal is precisely the aforementioned free oscillation ICW (which we can imagine as the precession of a small planet located inside the Earth's mantle following the analogy made in one of the references cited by the authors).

Therefore, my main suggestion is to make some changes in the wording of the article to show that not only the static inclination of the inner core axis has been determined, but also the value of the period of its free ICW motion has been determined for the first time experimentally, based on Earth rotation data.

I consider that the empirical determination of this value should be highlighted since in the last 30 years several values of the period, inferred from theoretical considerations, have been published (as cited in the references) and even in some recent work it has been obtained that the motion should occur in the retrograde direction and not in the prograde direction as most of the previous authors concluded (Chao, B. F. (2017), Dynamics of the inner core wobble under mantle-inner core gravitational interactions, *J. Geophys. Res. Solid Earth*, 122, doi:10.1002/2017JB014405). This shows how difficult is deriving accurate values of the ICW period from theory and geophysical models of the earth's interior. On the other hand, I would like to remind that the observed value of the Chandler period has not yet been fully justified, and that the most complete theoretical deductions often differ from the observed value by a few tens of days.

I think the authors could emphasize their experimental determination of the period without making major changes to the text or the abstract, although perhaps the title should be modified so as not to emphasize only the inclination. Regarding the magnitude of the difference with former estimations of the period, based on quite different methodologies because none uses accurately observed earth rotation data, I'd like to recall that reliable error bars are not available for any of the published determinations, given the difficulty of the implied calculations. That fact must be taken into account in any controversy that may be risen.

On the other hand, while the wording poses no problems for those with an intermediate knowledge of the earth's rotation, I think from my experience that a large number of potential readers interested in the subject do not reach that level. They are possibly well acquainted with the Chandler period arising from the existence of the liquid core, but rather less familiar with the two additional normal modes of rotation that appear when the solid earth model also contains an inner core. I therefore suggest that the authors consider adding a short paragraph describing the phenomenon.

Below, I suggest that the authors please consider making some minor editorial changes, but this suggestion does not mean that they should make them. They are (by lines):

L 21. Change "Earth"->"solid Earth".

L 26 Specify which theory is meant if the sentence refers to a specific one or make a more generic reference such as "most published theories...".

L 34-35 It would be useful to include some reference, either among those already cited or in addition to those already cited.

L 39 I think it would be useful to indicate that this fact will be demonstrated later, or to give a reference, for the convenience of readers.

L 54 It is actually in the caption of Figure 1, not in the figure itself.

L 55 The wording is not completely accurate, it is valid for the so-called Earth Rotation Parameters (ERP) which are the ones used in the derivation of the results, but not for the remaining two, namely precession-nutation ones.

L 79-80 Perhaps it would be useful to reinforce the reasoning made to reach the conclusion. Actually, it is not only possible to conclude, but it must be concluded because current theoretical knowledge offers no other possibility to explain the origin of this 8.5-year signal with the above-mentioned properties.

L 107-108 Although the sentence is true, it is perhaps worth expanding the explanation slightly.

L 191 Including a reference for the conventional value of Omega, among the many available sources

L 194 Rather than observation, one should say the motion of the observed pole (as a response).

L 208 Not necessarily in this line, but somewhere in the text or supplementary material, I think more details should be given on how the method has been applied, e.g., the size of the bins, whether the Q factor that appears as one of the input variables in the referenced routine (in L 228) has been given an infinite or finite value, etc., so as to ensure that the calculations are transparent and repeatable.

L 306-307 Correct the reference (Fong Chao, B...)

Reviewer #3 (Remarks to the Author):

Major points

1) Discussion of seismology

The discussion of the previous seismic work is not entirely reflective of what those studies found. I do not think it is fair to say that the studies are collectively arguing for a large static tilt of the inner core. Collectively, the cited studies (and others) have investigated the anisotropic and heterogeneous structure of the inner core, and whether there is differential axial rotation between the inner core and mantle. Some of the differential rotation studies (e.g. Wang & Vidale 2022, ref 10 of this work; and the recent article by Yang & Song 2023, Nature Geoscience) measure differential rotation by tracking/estimating seismic variations in the shallow inner core; these lateral variations provide no information on a potential tilt of the inner core. Studies of inner core anisotropy (e.g. refs 7 and 9 of this work) do lead to estimates of anisotropy axis that is tilted relative to the net rotation axis of the Earth. However, the physical mechanism responsible for that seismic anisotropy and, in particular, the physical meaning of the orientation of that fast axis are debated (even within individual observational studies), as noted in L118-120. The alignment of the fast axis from seismic anisotropy certainly need not be attributed to a tilt of the inner core. Furthermore, the review by Deuss (2014, ref 12 of this work) notes "The hemispherical variations removed the need for a tilted anisotropy axis and provide a better fit to the body wave data".

The inner core structure outlined by the recent study of Brett et al (2022, ref 29) and discussed on L130 have the western anisotropic zone confined to the northern hemisphere; I do not agree with the subsequent statement on L131 that this is indicative of a tilted figure axis, which should produce a comparable anomaly in the antipodal eastern + southern region of the inner core. Overall, I think the statements made in this work that a 5-10 degree tilt of the inner core figure axis are suggested by

seismology need to be removed as being unreflective of the overall message put forward by those studies.

2) Is 8.5 years the ICW and, if so, why?

As in their previous work (ref 16 of this work), the authors have found a number of peaks in polar motion. The 8.5 yr peak is argued to be the ICW as it shows in the prograde spectrum. The other peaks are of similar amplitude to the 8.5 yr signal and also show inconsistency between positive and negative PM spectra (e.g., the 7.3 yr signal is much larger in the negative half of figure 2a, the 5.9 year peak is considerably stronger in the positive half). Without understanding what those other peaks represent physically and why their amplitudes vary, it is not obvious that the 8.5 yr peak should be picked out as the ICW.

Let us accept that the 8.5 yr peak is the ICW. This is not the period predicted from previous theoretical models that account for Earth structure and match other observed wobbles and nutations (e.g., refs 1-6). Why does the period not match those models and predictions? This must tell us something about the structure of the inner core.

I grant that these are not the questions that the authors have focused on, but I think the existence of the other peaks and the mismatch between predicted and “observed” ICW periods are both points that are too significant to be left unexplored and unexplained.

Moderate points

- L99: How is the time series for the rate of change of ΔLOD constructed? I don't think it is provided directly by the data service; so, the authors have presumably calculated it. If so, what method was used and was the differencing done before/after filtering of ΔLOD ? Or did the authors calculate the derivative of their fit to the ΔLOD data? Such details would be useful for anyone trying to reproduce the work.

- Related to the above, it would be useful to have a specific data section in the Methods where the sources and full processing flows of each time series were clearly laid out rather than needing to pick the details out of the caption to supplementary figure 1.

- L108-110: I do not follow the argument here. What is “it” referring to in the phrase “...only when it is in the same plane...”? What do you mean by being in the same plane as the tilt angle? The axial torque vector Γ_z is always in the same plane as the vector defining the inner core figure axis, since any two (non-colinear) vectors from the origin define a plane.

•The Methods section shows how the equatorial and axial torques on the mantle are related to the changes in mantle rotation vector (observed polar motion and LOD signals); however, there is no explicit demonstration that the ratio of those torques equates to the tilt angle of the inner core. Given that the complete description of couple rotations and torques for the mantle-outer core-inner core system is far from trivial (particularly if we allow that the solid regions of the Earth may be somewhat triaxial), I think the authors should clearly articulate the reasoning here.

Minor points

- L21: Repetition of “solid inner core” one of which should be “solid mantle”.
- L35: “and etc.” is unnecessary.
- L67: “that has a significantly different feature” Different than what?
- L106: “using a more complicated method” is too vague. You don’t have to explain the method here but should at least be specific on what it was.
- L108- I would use “trough” rather than “valley”.
- L192- Am and Cm should have units of kg m^2 .

Responses to Reviewers

Reviewer #1 (Remarks to the Author):

A review of the paper: Inner core static tilt: Evidence from intradecadal oscillation in the Earth's rotation

by An et al.

This is a paper on the possibility that the inner core has a static tilt with the Earth's rotation axis.

Consider the very first sentence of your manuscript. “The Earth consists mainly of a solid inner core, a liquid outer core, and a solid inner core with the same center of mass.”

I don't know what the above sentence means, but the sloppiness, unfortunately, persists throughout the entire manuscript.

Response: Thanks for your so careful review, which help us to further improve our MS. First, we apologize for our carelessness, we did have some spelling mistakes (one of the ‘solid inner core’ should be changed to ‘solid mantle’; we have modified it, please see line 21). This time, we carefully re-checked the full text and made some adjustments. Besides, we have used a professional editing service (American Journal Experts) to polish the manuscript and asked a native speaker to help us to further improve the English. We hope the new MS will be much better.

Although geodynamics is not my area of expertise, and this paper needs to be assessed by someone with good knowledge of the current research on the tilt to judge the novelty you claim, you did not convince me that you can observe the tilt of the Earth's inner core, let alone prove it. There are two main reasons: firstly, in the abstract and elsewhere, you say that the tilt you infer is 0.17 degrees away from the Earth's spin axis. However, you do not present the uncertainty of that estimate, and given that it is probably quite large, it could be said that you obtained a 0.0-degree tilt or no tilt at all.

Response: Thanks for your constructive comments. According to your comments, we added the estimated uncertainties and explained how to estimate them. Here, the bootstrap procedure (Efron & Tibshirani, 1986) was used to evaluate the uncertainties, and a short explanation is given here:

1) We first chose the frequency-domain AR method to estimate the amplitude and period of the target ~ 8.5 yr spectral peak from the PM (1900-2020) and Δ LOD (1900-2020) records; the obtained estimated period/amplitude for the PM and Δ LOD are 8.52yr/4.7mas and 8.47yr/0.061ms,

respectively.

- 2) We chose the 0.09-0.105cpy and 0.12-0.15cpy as the background noise frequency bands, and the mean power of those two bands was chosen as the surrounding noise level R .
- 3) Taking the estimated 8.52yr signal with the amplitude of 4.7 mas as an example; we then synthesize a noise-free time series $f_0(t)$ containing a $1/8.52$ cpy frequency signal with an estimated complex amplitude of 4.7 mas. We also synthesize 300 white noise time series $f_i(t)$ ($i=1,2,\dots,300$) with the same noise level R (300 is sufficient statistically).
- 4) After putting $f_0(t)$ into the 300 noise time series $f_i(t)$, 300 new time series containing the target signal were obtained. Consequently, 300 new period/amplitude estimations for the target ~ 8.5 yr signal were obtained. Finally, the standard deviation of those 300 estimations is taken as the period (and amplitude) uncertainty.

Such process has been widely used for uncertainty estimations in different geophysical research (such as Earth's free oscillations (e.g., Widmer, et al., 1992; Häfner and Widmer, 2013; Ding and Chao, 2015) and the trend estimations from space geodetic observations (e.g., Chen et al., 2013; Han et al., 2017)).

The final estimated uncertainties are 0.19yr/0.4mas for the estimates from PM, and 0.32yr/0.007ms for the estimates from Δ LOD. The final estimate for the period can be obtained by weighting those period estimates, i.e., 8.5 ± 0.2 yr. Combining with the amplitude estimate 0.061 ± 0.007 ms (0.046 ± 0.005 ms/yr) in Δ LOD, we obtained that $|\Gamma_{\text{eq}}| = (2.87 \pm 0.24) \times 10^{19}$ Nm, and $|\Gamma_z| = (8.61 \pm 0.95) \times 10^{16}$ Nm, and according to the law of error propagation, we obtained that the static tilt angle $\theta = 0.17 \pm 0.03^\circ$. Given that the frequency-domain AR method has high precision for estimating the amplitude/period of a harmonic signal, we can see that our estimates are quite accurate, hence the uncertainty for the tilt is not so large. Hence, we tend to believe this static tilt should be reliable.

According to your comments, we added the corresponding uncertainties and expressed that we used a bootstrap procedure, and considering the paper-length limitation, we didn't explain the details. Please see lines 91 and 92 (*In addition, the uncertainties for the estimates in this study were based on a bootstrap procedure²⁹.*) and lines 89, 98, and 172, 176, 178, 181.

Reference:

Efron, B., & Tibshirani, R. (1986). Bootstrap methods for standard errors, confidence intervals, and

other measures of statistical accuracy. *Statistical Science*, 54-75.

Häfner, R., & Widmer-Schnidrig, R. (2013). Signature of 3-D density structure in spectra of the spheroidal free oscillation ${}_0S_2$. *Geophysical Journal International*, 192(1), 285-294.

Widmer, R., Masters, G., & Gilbert, F. (1992). Observably split multiplets—data analysis and interpretation in terms of large-scale aspherical structure. *Geophysical Journal International*, 111(3), 559-576.

Ding, H., & Chao, B. F. (2015). Detecting harmonic signals in a noisy time-series: the z-domain Autoregressive (AR-z) spectrum. *Geophysical Journal International*, 201(3), 1287-1296.

Chen, J. L., C. R. Wilson, and B. D. Tapley (2013), Contribution of ice sheet and mountain glacier melt to recent sea level rise, *Nat. Geosci.*, 6, 549–552, doi:10.1038/ngeo1829.

Han, J., N. Tangdamrongsub, C. Hwang, and H. Z. Abidin (2017), Intensified water storage loss by biomass burning in Kalimantan: Detection by GRACE, *J. Geophys. Res. Solid Earth*, 122, 2409–2430, doi:10.1002/2017JB01412

Secondly, a correlation is not causation. You will need more robust evidence that the inner core has a tilt apart from showing the two spectra and deducing that the signal you observe in the spectrum is the only candidate for the ICW. Take, as an example, a recent claim by seismologists that the rotation of the inner core has some correlation with the LOD change – no one in the seismological and the broad geophysical community can take it as proof that the two are related or caused by the same phenomenon due to a significant uncertainty of those estimates and a lack of physical mechanism explaining the correlation. This lack of uncertainty analysis and solid proof that you are observing a tilt of the inner core makes your paper highly speculative.

Response: Thanks for your constructive comments and suggestions. We fully agree with you that we cannot only use the related correlation to claim that the inner core must be static tilted. Until now, the only theoretical explanation that the ICW can also appear in ΔLOD is that there is a static tilt angle between the inner core's rotation axis and the mantle's rotation axis (see details at below). Without this static tilt angle, the ICW can only appear in the PM. At present, there is no other theoretical mechanism to explain the presence of the ICW in ΔLOD . Therefore, we only need to confirm that the ICW also appears in ΔLOD to verify that the inner core has a static tilt.

We have also noticed some recent seismological results about the rotation of the inner core, and

our corresponding author, Hao Ding, has participated in the discussion of some of the papers before their publication (such as Wang & Vidale, 2022 *SA*; Yang & Song, 2023, *NG*). Similar periods or similar waveforms are weaker constraints, for example, although the ~ 70 yr signal and the ~ 64 yr signal have similar waveforms in the time domain, they are very likely two different signals (the 18.6yr signal and the ~ 22.5 yr signals in the Δ LOD have the same situation). But the *same* period and good phase consistency are different cases, coincidence is possible, but it will be quite small. Even though, we still need to rule out that small probability.

We eliminate this small probability in our data processing. The Earth's rotation changes have two excited sources, one from the Earth's interior motions and another from the Earth's external mass changes. The latter includes three excitation sources, atmospheric press and non-tidal ocean and hydrological loading effects. Only when the external sources were ruled out can we be confident that the signal is caused by the internal sources. Taking the Δ LOD as an example, we have shown that atmospheric and oceanic effects have no significant contribution to the target frequency band (see Fig. S2a in the SI), here we further show the hydrological effects. Fig. R1 show the Fourier spectrum of the hydrological effects (HAM) of the Δ LOD. We can see that the amplitudes in the target frequency band are only ~ 0.005 ms, which is far less than the noise level of the observed Δ LOD in the same frequency band. Hence, we can also rule out the hydrological effects as the excited sources of the ~ 8.5 yr signal.

Figure R1. The Fourier spectrum of the hydrological effects (HAM) of the Δ LOD. The vertical dashed line denotes the 8.5yr.

In short, our recommendation was based on the following:

1) we ruled out the possibility of external excitation sources, and hence the 8.5yr must come from the Earth's interior motion;

2) the theoretical period of ICW was suggested in the range of 6.6-7.8yr (based on the PREM model, and different coupling effects were considered). This period is mainly dependent on the density jump $\Delta\rho_{ICB}$ at the inner core boundary, although the $\Delta\rho_{ICB}$ of the PREM was not particularly accurate, it will not deviate too much; if too much, the frequencies of other normal modes of the free oscillations which are sensitive to the core structures will have larger deviations with the theoretical values based on the PREM, but no such results were found. Hence, we can be sure that the period of the ICW should be close to 6.6-7.8 yr.

3) the ICW is a prograde motion that same as the well-known Chandler wobble; in the complex spectrum of the PM (with the complex from $x-i\cdot y$), a prograde wobble has a positive frequency and a retrograde wobble has a negative frequency; the Chandler wobble was just identified by that it is only present in the positive frequency axis of the PM's spectrum, the observed period of the CW is also ~ 30 days large than its theoretical value (Dumberry, 2009; Crossley & Rochester, 2014)). There are no other physical explanations for the signal based on current theoretical knowledge except the ICW. Hence, we determine that the 8.5yr signal must be the ICW.

4) the 8.5yr oscillation extracted from the $d\Delta LOD/dt$ is synchronized with the 8.5yr oscillation extracted from the y component of the PM, and has a fixed $\pi/2$ phase difference with that from the x component of the PM. This is not simply correlation; they have the same periods (considering estimate errors) and good phase consistency, and they are different components of the Earth's rotation.

Based on those, we can carefully suggest the 8.5yr oscillation in ΔLOD is also the ICW signal. As explained above, the only possibility is that there is a static tilt angle between the inner core's rotation axis and the mantle's rotation axis.

Figure R2, the diagram of the ICW

Next, we further explain why the inner core is tilted relative to the mantle can be inferred from the ICW present in the Δ LOD. The solid inner core and the mantle are separated by the liquid outer core, and thus inner core can move relative to the mantle. The ICW is the rotation eigenmode, and as shown in Figure R2, the small wobble of the inner core's figure axis around its rotation axis; this will exchange angular momentum with the mantle through the mantle-inner core gravitational torque Γ_{MICG} which changes only in the equatorial plane of the inner core. In fact, the ICW signal that appears in the PM requires a corresponding torque in the equatorial plane of the mantle, and then ICW signal that appears in the Δ LOD requires a corresponding torque in the direction of the mantle's rotation vector (vertical to the equatorial plane of the mantle). However, if the Earth is completely in the hydrostatic equilibrium, the inner core is not tilted relative to the mantle, that is, the equatorial plane of the inner core will coincide with that of the mantle, and the coupling torque Γ_{MICG} will change only in this coincident plane. Therefore, the torque Γ_{MICG} has no component in the rotation direction of the mantle, so it cannot change the rotation rate of the mantle, and the ICW signal will not appear in the Δ LOD. The only way is that there is a static tilt between the inner core and the mantle. Thus, the coupling torque Γ_{MICG} changes on the equatorial plane of the inner core and will produce a component in the rotation direction of the mantle, which will cause the corresponding change in the Δ LOD. Therefore, the ICW revealed in both the Δ LOD and the PM can be used to demonstrate the tilt of the inner core relative to the mantle. Besides, a tilted inner core will cause the ICW shown in the Δ LOD was also theoretically suggested by previous studies (Greiner-Mai et al. 2000).

According to your comments, we added some relevant expressions, please see lines 55-58: For the periodic signals present in the PM and Δ LOD, the consensus is that they were excited by the Earth's internal or external sources through the conversion of angular momentum¹⁷. Hence, we need to rule out the influence of external excitation sources before determining that a target signal is coming from the Earth's internal motion; and see lines 72-77: Different from the oceanic tidal signals that have both prograde and retrograde components in PM²²⁻²⁴, the ICW is a prograde motion (the same as the Chandler wobble, i.e., the mantle wobble); in the complex spectra of the PM, a prograde/retrograde wobble only has a positive/negative frequency. The identification of the well-known Chandler wobble is based on this feature²⁵⁻²⁸. Therefore, as a prograde motion, the ICW only appears on the positive frequency axis of the PM spectrum and this is a distinguishing feature for identifying it.

Reference:

- Crossley, D. J., & Rochester, M. G. (2014). A new description of Earth's wobble modes using Clairaut coordinates 2: Results and inferences on the core mode spectrum. *Geophysical Journal International*, 198(3), 1890-1905.
- Dumberry, M. (2009). Influence of elastic deformations on the inner core wobble. *Geophysical Journal International*, 178(1), 57-64.
- Greiner-Mai, H., Jochmann, H., & Barthelmes, F. (2000). Influence of possible inner-core motions on the polar motion and the gravity field. *Physics of the Earth and Planetary Interiors*, 117(1-4), 81-93.
- Wang, W., & Vidale, J. E. (2022). Seismological observation of Earth's oscillating inner core. *Science Advances*, 8(23), eabm9916.
- Yang, Y., & Song, X. (2023). Multidecadal variation of the Earth's inner-core rotation. *Nature Geoscience*, 16(2), 182-187.

Furthermore, you use some seismological papers that showed complexities in the inner core to reinforce that the inner core might have a non-axisymmetric mass, but in my opinion, a more rigorous understanding of what these papers are all about is lacking from the manuscript.

So, a tilt and a differential rotation of the inner core might be real as they emerge intuitively and numerically from geodynamical modeling (as shown in multiple papers), but you need more in this

paper to convince me that what you have achieved is a conceptual advance.

Response: Thanks for your very useful comments, we do have problems with accurate understanding and citation of these papers. Actually, as we have written in lines **118-120** in the old MS: *‘However, there is no evidence for the interpretation of the seismic anisotropy axis as the figure or rotation axis of the inner core and there is no consensus on what the anisotropy axis represents even in the mineral physics community’*. We found that some papers expressed that the seismic anisotropy axis was not the figure axis of the inner core, but we didn’t express them well in the old MS. Our initial idea was to state that such a perspective exists, without concern as to whether it has been falsified; because our understanding was that there were certain uncertainties about the inner core results from seismic observations, and are often controversial, so we cannot judge which one is right. Based on this comment and the following comments, we have changed the related expressions (more details will be given in the next response); we tend to emphasize that our results should be available as a constraint on the inner core structure retrieved from seismic observations, that is, the relevant seismological results should conform to our observations. Of course, such conformity can only be qualitative; although we are not experts in seismology, we believe that such a small tilt should be difficult to identify directly from seismic observations.

The related seismological expressions in lines 115-130 in the old MS were mainly used to support the idea that the inner core might be tilted to the west. In fact, we don't need to refer to the anisotropic axis results from seismology (of course, as you commented, they are different things). Ever since the possibility of a static tilt of the inner core was proposed for explaining the decadal variations in the PM might be the ICW, it has been assumed that it should be in line with the dipole axis of the geomagnetic field, i.e., 10° away from the rotation axis of the mantle (e.g., Schmutzer, 1978; Szeto and Smylie, 1984, 1989; Greiner-Mai, 1997; Greiner-Mai et al. 2000). Guo & Ning (2002) followed this suggestion, they simply considered that the inner core and mantle are rigid and axially symmetrical, and the mantle is rotating uniformly in the space, then they found that a 10° tilt of the inner core will increase the ICW period by about 70 days. Although the decadal variations in the PM previously proposed have been confirmed cannot be the ICW, the possibility of a westward tilted inner core was not denied. Besides, the phase synchronization of the ~ 8.5 yr oscillations in the $d\Delta LOD/dt$ and in the y component of the PM can directly infer that the inner core should be tilted to the west, because the y component of the PM along the $90^\circ W$ longitude. Hence, combing with your comments, we changed

the related expressions as: “To explain the decadal oscillations in both the PM and Δ LOD as the possible ICW, the rotation axis Ω'_m of the inner core was thought to no longer be the common axis Ω_m but have a static tilt relative to the mantle, i.e., the inner core rotation axis coincides with the dipole axis of the geomagnetic field (tilted 10° westwards from Ω_m)⁸. Although there is still no confirmed observation of the ICW^{9,10}, a static tilted inner core remains possible, and the effect of a 10° static tilt on the ICW period was further investigated⁹.” (please see lines **30-35**); and “Given the y component of the PM along the 90° W longitude, the phase synchronization in Figures 3a and 3c indicates that the inner core is more likely tilted in the 90° W direction, which is also similar to that suggested by previous studies^{8,9,31,32}.” (please see lines **128-130**).

Except for the above-mentioned reference, few other studies have specifically considered a static tilted inner core (e.g., Mathews et al., 1991; 2002; Rochester & Crossley, 2009; Rochester et al., 2014; Dumberry, 2009), the ‘tilt’ they called is just the ‘dynamic tilt’ that we explained in lines **26-28**. Different from previous studies, we further considered an ellipsoidal inner core, and under the core-mantle coupling mechanism, we derived for the first time the variations of Δ LOD caused by the ICW when the inner core has a static tilt. In addition, none of Greiner-Mai et al. and Guo et al.’s formulas can directly infer the static tilted angle of the inner core from the ICW signal in the Δ LOD and PM.

Therefore, in contrast to previous studies, we believe that we are still taking a certain conceptual advance -- of course, we do not want to emphasize a new concept from scratch, we just want to confirm an existing but doubtful concept. From the literature available so far, we tend to think that we have indeed found strong evidence for the existence of this static tilt for the first time. According to your comments, we added a new note in lines **223-225**: “Combing with Eqs. (1), (2), (6) and (7), we can directly infer the static tilted angle of the inner core from the ICW signal in the Δ LOD and PM, which is impossible in related previous studies^{8,9}.”

Reference:

- Dumberry, M. (2009). Influence of elastic deformations on the inner core wobble. *Geophysical Journal International*, 178(1), 57-64.
- Greiner-Mai, H. (1997). Possible relations between the rotational axis of the earth's inner core and the magnetic dipole axis. *Astronomische Nachrichten*, 318(1), 63-71.
- Greiner-Mai, H., Jochmann, H., & Barthelmes, F. (2000). Influence of possible inner-core motions on

the polar motion and the gravity field. *Physics of the Earth and Planetary Interiors*, 117(1-4), 81-93.

Guo, J. Y., & Ning, J. S. (2002). Influence of inner core rotation and obliquity on the inner core wobble and the free inner core nutation. *Geophysical Research Letters*, 29(8), 45-1.

Mathews, P. M., Buffett, B. A., Herring, T. A., & Shapiro, I. I. (1991). Forced nutations of the Earth: Influence of inner core dynamics: 1. Theory. *Journal of Geophysical Research: Solid Earth*, 96(B5), 8219-8242.

Mathews, P. M., Herring, T. A., & Buffett, B. A. (2002). Modeling of nutation and precession: New nutation series for nonrigid Earth and insights into the Earth's interior. *Journal of Geophysical Research: Solid Earth*, 107(B4), ETG-3.

Rochester, M. G., & Crossley, D. J. (2009). Earth's long-period wobbles: a Lagrangean description of the Liouville equations. *Geophysical Journal International*, 176(1), 40-62.

Rochester, M. G., Crossley, D. J., & Zhang, Y. L. (2014). A new description of Earth's wobble modes using Clairaut coordinates: 1. Theory. *Geophysical Journal International*, 198(3), 1848-1877.

Schmutzer, E. (1978). Investigation on the influence of the global magnetic field of the Earth on the motion of the solid core.(Declination, westward drift, northward drift etc.). *Gerlands Beitrage zur Geophysik*, 87(6), 455-468.

Szeto, A. M. K., & Smylie, D. E. (1984). Coupled motions of the inner core and possible geomagnetic implications. *Physics of the Earth and Planetary Interiors*, 36(1), 27-42.

Szeto, A. M. K., & Smylie, D. E. (1989). Motions of the inner core and mantle coupled via mutual gravitation: regular precessional modes. *Physics of the Earth and Planetary Interiors*, 54(1-2), 38-49.

Let me comment further on your reference to the seismological papers.

You misunderstood the goal of most inner-core seismological papers you cite. The goal was not to establish a tilt of the inner core but to study anisotropy. Then, by making a strong assumption that anisotropy could be approximated by a cylindrical model, one can perform a grid search to see if the data one has would point to the main axis of anisotropy oriented in a slightly different direction than the spin axis of the Earth. You can then obtain some mean, which cannot be oriented exactly along the Earth's rotation axis because of the noise in the traveltime data. What's worse than that is that we don't

have an even volumetric sampling of the ray paths sensitive to the inner core. You cite the paper by Su et al. (1996) and their estimate of the tilt within the range of 5–10 degrees, but you are perhaps unaware that this tilt was shown to be meaningless by Souriau et al. (1997) because of the bias in the data. By the way, they didn't find any wobble in time, even with that biased estimate. The bottom line is that these early estimates of the anisotropy axis were shown to be unreliable and erroneous. For that reason, they were abandoned, and the differential rotation of the inner core estimates were conducted by other methods. Therefore, your underlying thesis that your estimate of the tilt is two orders of magnitude lower than what is obtained by seismology is wrong and misleading. Seismology has not yet determined the tilt of the inner core or anything like that.

Response: Thanks a lot for your detailed explanation, and we have benefited a lot. As we responded above, we were indeed aware that Su et al. (1996) 's results were not considered to be the inner core static tilted; as you said, Su et al. (1996) 's results have been proven wrong by Souriau et al. (1997). We also don't think that seismic observations can find the ICW signals directly.

In the old MS, we referred to an 8° tilted inner core from seismological results, this was based Wang & Vidale (2022, *EPSL*). Applying a novel back-projection method, Wang & Vidale (2022, *EPSL*) locate the inner-core regions that scatter the energy within the *PKiKP* coda based on its slowness and the lapse time. By relaxing the constraint that the inner core rotation axis is the same as the Earth's rotation axis, Wang & Vidale (2022 *EPSL*) grid-search the tilt of the rotation axis, the azimuth of the tilt, and the rotation rate to find the best match to the time shifts of the high-energy points. The best-fitting result indicates a rotation axis that is 8° off the geographic (mantle) north pole. However, Wang & Vidale (2022 *EPSL*) denotes that this tilted axis is only a marginally better fit to the observed time shifts, and the seismic waves lack resolution for the tilt of the inner core axis. After discussing with their first author (W. Wang), and combining with your comments, we decided not to show this again; and as we responded above, we directly refer to the theoretical suggestions given by geodynamical studies.

The differential rotation of the inner core and the ICW are two different concepts, the former mainly refers to the change of the inner core rotation rate, while the latter refers to the change of the inner core's pole, which are similar to the Earth's Δ LOD and PM (they are also two different concepts).

According to your suggestions, in order to avoid misleading readers, we have deleted those related expressions and made corresponding modifications (please see lines 30-35 and 130-142).

Reference:

- Souriau, A., Roudil, P., & Moynot, B. (1997). Inner core differential rotation: facts and artefacts. *Geophysical Research Letters*, 24(16), 2103-2106.
- Su, W. J., Dziewonski, A. M., & Jeanloz, R. (1996). Planet within a planet: rotation of the inner core of Earth. *Science*, 274(5294), 1883-1887.
- Wang, W., & Vidale, J. E. (2022). Earth's inner core rotation, 1971 to 1974, illuminated by inner-core scattered waves. *Earth and Planetary Science Letters*, 577, 117214.
-

Also, let me comment on your invoking seismology as a method that found that the mass distribution in the Earth's inner core is not axisymmetric. While heterogeneities are documented to exist near the inner core boundary, everything beyond that is much more speculative. If we accept that one hemisphere of the inner core is faster than the other, would that necessarily indicate a denser material and more mass? Why would anisotropic fabric be a denser fabric? The opposite can be true. From seismology, anisotropy appears weaker in the outer part of the inner core and stronger towards its center. It does not necessarily have anything to do with cylindrical anisotropy's fast and slow axis but with the pronounced slow axis in the innermost part of the inner core. For instance, deeper into the inner core, two recent studies that you cited show an offset of the anisotropic (as hypothesized) volume from the center at two different locations of the inner core. In all honesty, there is no consensus as to which one is correct, so your inference that a tilt is to the western side needs more seismology support. But if you honestly present seismological work as it is, you will find it more difficult to substantiate your claims.

Response: Thanks for your advice and we learn a lot. Indeed, as you say, if only one hemisphere is faster than the other, it is difficult to confirm whether the density is denser or not, which is similar to the controversy about the Large Low Shear Velocity Provinces (LLSVPs).

In the old MS, we weren't focused enough, so there was some ambiguity. We have deleted some unnecessary discussion about some literature this time, and mainly focus on three relevant papers (Alboussière et al. (2010 *Nature*), Monnereau et al. (2010 *Science*) and Zhang et al. (2019 *JGR-SE*)).

In the caption of Figure 3 of Monnereau et al. (2010 *Science*) (also Figure R3), they noted that: "In a superadiabatic regime, any thermal heterogeneity of harmonic degree-one shifts the center of

mass of the inner core toward its colder and denser hemisphere (left). The equilibrium position at the center of mass of Earth (o) is restored by a translation of the inner core (from dashed to solid positions). This induces a topography h that is not in equilibrium with the pressure and temperature conditions within the outer core. ***Crystallization on the denser hemisphere (left)*** and melting on the opposite side act to remove the topography, but in return amplify the density heterogeneity and maintain the center of mass shifted toward the crystallizing side....”. Although they wanted to explain a possible inner translational mechanism, they did discuss the possibility that the inner core was denser in the western hemisphere. This study is related to Alboussière et al. (2010, *Nature*), which discussed a uniform flow caused by the crystallization and melting at the surface of the inner core, the associated changes in density and gravitational potential lead to the inner core has an eastward translation along the x -axis. In Alboussière et al. (2010)’s dynamical model, the western hemisphere of the inner core is denser than its eastern hemisphere (see their Fig. 4). In addition, Zhang et al. (2019 *JGR-SE*) concluded that “Earth’s western hemisphere has a thicker compacting layer (F’ layer) than the eastern hemisphere on average” based on the significant difference in seismic velocity between the eastern and western hemispheres, this may also mean denser material in the western hemisphere.

As we have responded above, we don't really need seismological results to tell us that the inner core's static tilt is westward. However, it is possible that these seismological results are at least qualitatively consistent with ours (denser material in the northwest hemisphere, even may only be in the top layer of the inner core, may cause the inner core’s figure axis tilts to the west).

Considering your comments, we modified the related expressions, please see lines 133-142: “Considering the crystallization and melting at the surface of the inner core, a previous dynamical model has similar suggestions³³, i.e., the western hemisphere of the inner core is denser than its eastern hemisphere. Interestingly, although it has some uncertainties³⁴, a seismological study has suggested that the western hemisphere of the inner core may be relatively denser³⁵ due to its lopsided growth based on different seismic anisotropy between its eastern and western hemispheres^{36,37}, and a thicker compacting layer at the top of the inner core's western hemisphere was also suggested³⁸; the more nuanced researches also revealed the differences in latitude distributions of those anisotropy regional structures³⁹ and found that the western anisotropic zone is largely confined to the northern hemisphere⁴⁰; those suggestions are generally consistent with the westwards statically tilted inner core that we found.”. In addition, we further modified the related discussion parts, we emphasized

qualitative analysis of the possible correlation between the westward tilted inner core we found and the seismological results, rather than using the seismological results as a prerequisite (because it is not necessary); please see lines 196-202: “Undeniably, it is difficult for seismological observations to detect such inner core static tilt directly, but interestingly, the results from seismological studies showed that the western/northwestern hemisphere (or at least its top layer) of the inner core may be relatively denser^{34,35,38,40}. These suggestions, although they have some uncertainties, are qualitatively consistent with our finding of a westwards-tilted inner core. Therefore, we also suggest that the static tilted inner core can be used as a qualitative constraint on the inner core structure retrieved from seismic observations.”.

[Redacted]

Figure R3. From Monnereau et al. (2010; *Science*).

[Redacted]

Figure R4. From Zhang et al. (2019 *JGR-SE*)

Reference:

Alboussière, T., Deguen, R., & Melzani, M. (2010). Melting-induced stratification above the Earth's inner core due to convective translation. *Nature*, 466(7307), 744-747.

Monnereau, M., Calvet, M., Margerin, L., & Souriau, A. (2010). Lopsided growth of Earth's inner core. *Science*, 328(5981), 1014-1017.

Zhang, Y., Nelson, P., Dygert, N., & Lin, J. F. (2019). Fe alloy slurry and a compacting cumulate pile across Earth's inner-core boundary. *Journal of Geophysical Research: Solid Earth*, 124(11), 10954-10967.

There is no such thing as “super-rotation”. (Line 34) We talk about the differential rotation of the inner core relative to the mantle. Besides, the paper would benefit from a discussion how the inner core tilt would impact the differential rotation studies.

Response: Thanks for your comments. We also prefer to believe the inner core should be differential rotation but not ‘super-rotation’, in particular, recent results suggest that the inner core is more likely to oscillate. The ‘super-rotation’ also has appeared in many previous literature, even on Prof. Song’s

homepage (<https://xiaodongsong.web.illinois.edu/news/>), in which, they highlighted that “2005 Confirmation of the inner core *super-rotation*”.

According to your suggestion, we changed it to ‘differential rotation’ (please see line 41). As for “*how the inner core tilt would impact the differential rotation studies*”, quantitative interpretation is not an easy task, and it is difficult for us to adequately and accurately state it in this study. Qualitatively, however, we know that the eastward differential rotation rate of the inner core will shorten the period of the ICW. Hence, we only added a note: “The larger observed period may also indicate that the eastwards differential rotation rate of the inner core should be much less than 1° per year^{9,14}.” (please see lines 193-195).

The quasi-eastern and quasi-western anisotropy is not that established. The robust differences are documented in absolute velocity between the eastern and the western hemisphere, but only at the top of the inner core. Given the sparse coverage of the inner core on the western side, whether the same holds for anisotropy is hard to tell. As you go deeper toward the center, the hemispherical division loses its meaning, and there is no consensus that any hemispherical pattern exists.

Response: Thanks for your useful comments and kind reminder. As we responded above, we only want to qualitatively state that some of the seismological results are likely to be consistent with the westward tilt we have obtained, and in the new manuscript, we do not want to use the seismological results as a prerequisite for determining the specific mechanism of our results. In addition, it is not necessary to go deep into the inner core; the outermost uneven density has the greatest effect on the figure axis of the inner core because the moment of inertia (defining the figure axis) is the integral of the density and the square of the distance over the body. Therefore, assuming that the anomalies are only at the top of the core, they should be considered consistent with our findings as long as they are likely to indicate that the average density in the eastern and western hemispheres of the inner core is uneven. In the new manuscript, we have changed to a qualitative description of this possible correlation (please see lines 133-142 and 196-202).

Finally, we sincerely thank you for your constructive suggestions and comments. In the old MS, we over-emphasized the results of the tilt determined by seismology, which led to some confusion in the discussion. In the new MS, we made a logical adjustment; we emphasize that the tilt is determined by the Earth's rotation, and it is most likely caused by the western hemisphere of the inner core having

a greater average density, which may be consistent with some seismological results (but still waiting for future seismological verification). We hope you will find our reply satisfactory. Thank you!

#####

Reviewer #2 (Remarks to the Author):

First of all, I would like to state for the record that I believe that the authors have clearly and convincingly demonstrated that the results they present in the article are true. Moreover, I would like to add that in my view they are truly important and sound. Applying a methodology that has already shown its power in other applications to geodetic problems and using precise earth rotation data, they have proved that the axis of the earth's solid inner core is tilted with respect to the polar axis of the whole earth, and at the same time they have determined for the first time empirically the value of the so-called ICW period. This motion can be thought of as analogous to the precession motion of the earth under lunisolar attraction, but in this case it is the inner core under the attraction of the earth's mantle surrounding it. This free mode of the Earth's rotation has so far only been known from theoretical developments and the value of its period inferred from geophysical models, as well as more indirectly from its effects on the Earth's rotation, more precisely on nutation, since the introduction in 1991 of a basic earth model made of 3 ellipsoidal, concentric and aligned layers, which was later refined and fitted to VLBI data, becoming the official IAU2000 nutation model.

Response: Dear Prof. Ferrandiz,

We appreciate your recognition and encouragement of our research, which gives us the confidence and motivation to further revise and improve our MS. We carefully addressed each of your comments and updated the manuscript accordingly. In the following, we listed detailed responses and made corrections carefully based on your comments and suggestions.

Although the authors only highlight the importance of the experimental determination of the inner core inclination from earth rotation data, more precisely from pole motion (PM) and length-of-day (LOD), I consider that the determination of the value of this ICW period is not less important than the previous finding and I could even dare to say that for me it could be even more important in that the accuracy of the method they use to identify weak harmonic signals (stabilized AR-z spectrum) is very precise in determining the value of the periods, but does not allow determining the amplitudes with

the same relative accuracy as far as I know.

Response: Thank you very much for the thoughtful review and encouragement to help improve this study. We further modified the related parts according to your insightful comments and suggestions, and to emphasize the importance of the observed ICW, we also further apply this observed period to constrain density jumps at the ICB, which is also a very important parameter but still controversial (please see lines 183-189, 195 and 196, and 239-255).

In my opinion, there is no doubt about the veracity of the results, since they clearly identify a harmonic constituent of the pole motion with a period close to 8.5 years, formed only by the prograde component, and also an oscillation of smaller amplitude in LOD with the same period; current theoretical knowledge can only explain this coincidence if the origin of the signal is precisely the aforementioned free oscillation ICW (which we can imagine as the precession of a small planet located inside the Earth's mantle following the analogy made in one of the references cited by the authors). Therefore, my main suggestion is to make some changes in the wording of the article to show that not only the static inclination of the inner core axis has been determined, but also the value of the period of its free ICW motion has been determined for the first time experimentally, based on Earth rotation data.

Response: Thanks for your very useful comments! “*which we can imagine as the precession of a small planet located inside the Earth's mantle*” is a very revealing idea, the phenomenon we want to explain is indeed similar to this. According to your suggestion, we further enhanced the importance of the observed 8.5yr period. Please see lines 190 and 191: “based on the Earth's rotation observations (PM and Δ LOD), we experimentally confirmed for the first time that the 8.5yr signal is the ICW.”

I consider that the empirical determination of this value should be highlighted since in the last 30 years several values of the period, inferred from theoretical considerations, have been published (as cited in the references) and even in some recent work it has been obtained that the motion should occur in the retrograde direction and not in the prograde direction as most of the previous authors concluded (Chao, B. F. (2017), Dynamics of the inner core wobble under mantle-inner core gravitational interactions, J. Geophys. Res. Solid Earth, 122, doi:10.1002/2017JB014405). This shows how difficult is deriving accurate values of the ICW period from theory and geophysical models of the earth's

interior. On the other hand, I would like to remind that the observed value of the Chandler period has not yet been fully justified, and that the most complete theoretical deductions often differ from the observed value by a few tens of days.

Response: Thanks for your very useful comments. As you noted, even the theoretical period of the ICW is quite difficult to determine, a small oversight may lead to a wrong result. I (H. Ding) participated in the discussion of Chao (2017)'s draft, but didn't find that the used magnitude of q_{20}^M has a mistake; after it was published, Rochester and Crossley found this and co-authored with Ben Chao, they published a Correction: *Rochester, M. G., Crossley, D. J., & Chao, B. F. (2018). On the physics of the inner core wobble; corrections to "Dynamics of the inner-core wobble under mantle-inner-core gravitational interactions" by B. F. Chao. Journal of Geophysical Research: Solid Earth, 123, 9998–10,002.*". In which, they confirmed that the theoretical period of the ICW based on the PREM model is still a $\sim+7$ yr prograde motion, not a 15yr retrograde motion.

Your comments about the Chandler wobble will help the readers to further understand the ICW, so we added some new discussions about it. Please see lines 183-186: "Our observed ICW period is slightly larger than the theoretical values (6.6-7.8yr)^{1,5-7}, but considering that even the generally accepted Chandler wobble observation period of ~ 430 days is ~ 30 days longer than its theoretical periods and that the density jump $\Delta\rho_{ICB}$ at the ICB was also poorly determined, this deviation is accepted."

I think the authors could emphasize their experimental determination of the period without making major changes to the text or the abstract, although perhaps the title should be modified so as not to emphasize only the inclination. Regarding the magnitude of the difference with former estimations of the period, based on quite different methodologies because none uses accurately observed earth rotation data, I'd like to recall that reliable error bars are not available for any of the published determinations, given the difficulty of the implied calculations. That fact must be taken into account in any controversy that may be risen.

Response: Thank you very much for your understanding that we did not give the uncertainty estimates in the old MS. As the other reviewers have pointed out that we didn't give the uncertainties, hence, in this version, based on the used PM and ΔLOD records, we further estimated the uncertainties of the period and amplitudes of this ~ 8.5 yr signal.

On the other hand, while the wording poses no problems for those with an intermediate knowledge of the earth's rotation, I think from my experience that a large number of potential readers interested in the subject do not reach that level. They are possibly well acquainted with the Chandler period arising from the existence of the liquid core, but rather less familiar with the two additional normal modes of rotation that appear when the solid earth model also contains an inner core. I therefore suggest that the authors consider adding a short paragraph describing the phenomenon.

Response: Thanks for your useful suggestions. In the new MS, we added some expressions, please see lines 26-35, and 72-77.

Below, I suggest that the authors please consider making some minor editorial changes, but this suggestion does not mean that they should make them. They are (by lines):

Response: Thanks for your careful review. We modified them as you suggested. Please see below.

L 21. Change "Earth"->"solid Earth".

Response: Done. Please see line 21.

L 26 Specify which theory is meant if the sentence refers to a specific one or make a more generic reference such as "most published theories...".

Response: Thanks for your suggestion! To more clearly explain the “static tilt”, we have rewritten this sentence as “The above ‘tilt’ is a generally dynamic tilt, and in this case, the ICW theoretically appears only in the polar motion (PM) of the Earth's rotation^{2,4}.”. Please see lines 26-28.

L 34-35 It would be useful to include some reference, either among those already cited or in addition to those already cited.

Response: Done as you suggested. Please see line 41.

L 39 I think it would be useful to indicate that this fact will be demonstrated later, or to give a reference, for the convenience of readers.

Response: Thanks for your kind reminder! We added some references for this. Please see line 44.

L 54 It is actually in the caption of Figure 1, not in the figure itself.

Response: Thanks for your kind reminder! Combing with the suggestions given by reviewer #3, we moved the pretreatments into the Method parts. Please see lines 54, 257-274.

L 55 The wording is not completely accurate, it is valid for the so-called Earth Rotation Parameters (ERP) which are the ones used in the derivation of the results, but not for the remaining two, namely precession-nutation ones.

Response: Thanks for your suggestions! We changed it to “the PM and Δ LOD changes”. Please see lines 58 and 59.

L 79-80 Perhaps it would be useful to reinforce the reasoning made to reach the conclusion. Actually, it is not only possible to conclude, but it must be concluded because current theoretical knowledge offers no other possibility to explain the origin of this 8.5-year signal with the above-mentioned properties.

Response: This is a very useful suggestion! According to this, we further modified this part, please see lines 89 and 90: “Since there is no other mechanism has been proposed to account for such a prograde \sim 8.5yr motion, and...”.

L 107-108 Although the sentence is true, it is perhaps worth expanding the explanation slightly.

Response: Thanks for your reminder! Here we added a note to suggest the reader see Figure 4 (please see lines 120 and 121).

L 191 Including a reference for the conventional value of Omega, among the many available sources

Response: Done as your suggestion! Please see line 215.

L 194 Rather than observation, one should say the motion of the observed pole (as a response).

Response: Done as you suggested. Please see lines 218 and 219.

L 208 Not necessarily in this line, but somewhere in the text or supplementary material, I think more

details should be given on how the method has been applied, e.g., the size of the bins, whether the Q factor that appears as one of the input variables in the referenced routine (in L 228) has been given an infinite or finite value, etc., so as to ensure that the calculations are transparent and repeatable.

Response: Thanks for your suggestion. We modified this part and added some notes in the SI. Please see lines 233-237, and the SI.

L 306-307 Correct the reference (Fong Chao, B...)

Response: Corrected. Please see line 379.

Sincerely,

Jose M Ferrandiz

#####

Reviewer #3 (Remarks to the Author):

Major points

1) Discussion of seismology

The discussion of the previous seismic work is not entirely reflective of what those studies found. I do not think it is fair to say that the studies are collectively arguing for a large static tilt of the inner core. Collectively, the cited studies (and others) have investigated the anisotropic and heterogeneous structure of the inner core, and whether there is differential axial rotation between the inner core and mantle. Some of the differential rotation studies (e.g. Wang & Vidale 2022, ref 10 of this work; and the recent article by Yang & Song 2023, Nature Geoscience) measure differential rotation by tracking/estimating seismic variations in the shallow inner core; these lateral variations provide no information on a potential tilt of the inner core. Studies of inner core anisotropy (e.g. refs 7 and 9 of this work) do lead to estimates of anisotropy axis that is tilted relative to the net rotation axis of the Earth. However, the physical mechanism responsible for that seismic anisotropy and, in particular, the physical meaning of the orientation of that fast axis are debated (even within individual observational studies), as noted in L118-120. The alignment of the fast axis from seismic anisotropy certainly need not be attributed to a tilt of the inner core. Furthermore, the review by Deuss (2014, ref 12 of this work) notes “The hemispherical variations removed the need for a tilted anisotropy axis and provide a better

fit to the body wave data”.

The inner core structure outlined by the recent study of Brett et al (2022, ref 29) and discussed on L130 have the western anisotropic zone confined to the northern hemisphere; I do not agree with the subsequent statement on L131 that this is indicative of a tilted figure axis, which should produce a comparable anomaly in the antipodal eastern + southern region of the inner core. Overall, I think the statements made in this work that a 5-10 degree tilt of the inner core figure axis are suggested by seismology need to be removed as being unreflective of the overall message put forward by those studies.

Response: Thank you for your very useful and constructive suggestions and comments. As you commented, our discussion about the anisotropic axis being the inner core's static tilt axis was not appropriate. In the new MS, we have revised this, and we no longer emphasized the seismological results about the 5-10° anisotropic axis (namely, we have deleted those parts as your suggestion); the existing geodynamic literature has directly indicated that the inner core may have a westward tilt of 10°. Hence, according to your comments, we have modified some related expressions, such as line **14** and **15**: “which is two orders of magnitude lower than that 10° assumed in geodynamics.”, and lines **30-35**: “To explain the decadal oscillations in both the PM and Δ LOD as the possible ICW, the rotation axis Ω'_m of the inner core was thought to no longer be the common axis Ω_m but have a static tilt relative to the mantle, i.e., the inner core rotation axis coincides with the dipole axis of the geomagnetic field (tilted 10° westwards from Ω_m)⁸. Although there is still no confirmed observation of the ICW^{9,10}, a static tilted inner core remains possible, and the effect of a 10° static tilt on the ICW period was further investigated⁹.”.

As for the related expressions in lines **115-132** in the old MS, our primary intention was to further illustrate from the seismological results that the 90°E-90°W tilt of the inner core should be more specifically westward. But as you commented, this should not be appropriate. In fact, we don't need to refer to the anisotropic axis results from seismology. Since the possibility of a static tilt of the inner core was proposed for explaining the decadal variations in the PM might be the ICW, it has been assumed that it should be in line with the dipole axis of the geomagnetic field, i.e., westward 10° away from the rotation axis of the mantle (Greiner-Mai et al. 2000). Although the decadal variations in the PM have been confirmed cannot be the ICW (Guo et al., 2005), the possibility of a westward tilted inner core was not denied. Besides, the phase synchronization of the ~8.5yr oscillations in the

$d\Delta LOD/dt$ and in the y component of the PM can directly infer that the inner core should be tilted to the west, because the y component of the PM along the $90^\circ W$ longitude. Thus, considering your suggestion, we find that the logic in which we discussed seismological results needs to be changed, not to treat them as prerequisites, but as a possible correlation. Hence, we changed the related expressions (lines **115-132** in the old MS) as: “Given the y component of the PM along the $90^\circ W$ longitude, the phase synchronization in Figures 3a and 3c indicates that the inner core is more likely tilted in the $90^\circ W$ direction, which is also similar to that suggested by previous studies^{8,9,31,32}. In terms of the long-term dynamic conservation of the Earth's angular momentum, this static westwards tilt is most likely due to the effect of the non-axisymmetric mass of the inner core, i.e., the western hemisphere (more specifically, the northwestern hemisphere) of the inner core should have greater average densities. Considering the crystallization and melting at the surface of the inner core, a previous dynamical model has similar suggestions³³, i.e., the western hemisphere of the inner core is denser than its eastern hemisphere. Interestingly, although it has some uncertainties³⁴, a seismological study has suggested that the western hemisphere of the inner core may be relatively denser³⁵ due to its lopsided growth based on different seismic anisotropy between its eastern and western hemispheres^{36,37}, and a thicker compacting layer at the top of the inner core's western hemisphere was also suggested³⁸; the more nuanced researches also revealed the differences in latitude distributions of those anisotropy regional structures³⁹ and found that the western anisotropic zone is largely confined to the northern hemisphere⁴⁰; those suggestions are generally consistent with the westwards statically tilted inner core that we found.” (please see lines **128-142**). As you can see, the expression of the seismological results has been changed to qualitative possible correlations.

As for your comment: “*which should produce a comparable anomaly in the antipodal eastern + southern region of the inner core*”, we fully agree with you, this is indeed a possible scenario. This time, we only express a simpler scenario, i.e., mass anomaly only in the northwest hemisphere of the inner core also causes a similar tilt westward, and we show this result in Figure R5a. Focusing on the figure axis (defined by the direction of maximum moment of inertia), the slightly flattened inner core can be viewed as the sum of a standard large sphere and two small spheres at both poles (north and south). The large sphere can be removed due to the axisymmetric mass, and only the two small spheres are shown in Figure R5a. We can see that the C -axis figure axis points in a north-south direction. With an identical anomaly in the western + northern hemisphere (i.e., add an identical small sphere to the

western + northern hemisphere), as shown in Figure R5b, we can clearly see that the center of mass shifts slightly north to the west and the figure axis of the core to shift westward. Thus, only anomaly in the western + northern hemisphere of the inner core also causes a tilt westward. In fact, the anomaly is small, the inner core tilt and the northward shift of the center of mass are small, and it is mainly a slow translational motion eastward of the inner core caused by the lopsided growth of the inner core in the western hemisphere and eastern hemisphere under the influence of gravity (Monnereau et al., 2010).

Figure R5. Simplified oblate sphere model, the axisymmetric sphere has been removed. (a) A standard oblate sphere is simplified to two small spheres at the north and south poles; the center of mass O is in the center, and the figure C -axis points in a north-south direction. (b) Add the same sphere (an identical anomaly) to the western + northern hemisphere, and the center of mass shifts slightly north to the west and the figure axis of the inner core shifts westward.

In addition, we tend to believe that the inner core static tilt cannot be detected *directly* by tracking/estimating seismic variations in the shallow inner core (as you have pointed out); because even the wobble angle of the ICW itself is hard to be detected in this way. Instead, the inner core oscillation may be detected in this way; the inner core oscillation is quite different from the ICW, the former is related to the inner core rotation speed while the latter is related to the pole's motion of the inner core.

Reference:

Greiner-Mai, H., Jochmann, H., & Barthelmes, F. (2000). Influence of possible inner-core motions on

the polar motion and the gravity field. *Physics of the Earth and Planetary Interiors*, 117(1-4), 81-93.

Guo, J. Y., & Ning, J. S. (2002). Influence of inner core rotation and obliquity on the inner core wobble and the free inner core nutation. *Geophysical Research Letters*, 29(8), 45-1.

Monnereau, M., Calvet, M., Margerin, L., & Souriau, A. (2010). Lopsided growth of Earth's inner core. *Science*, 328(5981), 1014-1017.

2) Is 8.5 years the ICW and, if so, why?

As in their previous work (ref 16 of this work), the authors have found a number of peaks in polar motion. The 8.5 yr peak is argued to be the ICW as it shows in the prograde spectrum. The other peaks are of similar amplitude to the 8.5 yr signal and also show inconsistency between positive and negative PM spectra (e.g., the 7.3 yr signal is much larger in the negative half of figure 2a, the 5.9 year peak is considerably stronger in the positive half). Without understanding what those other peaks represent physically and why their amplitudes vary, it is not obvious that the 8.5 yr peak should be picked out as the ICW.

Response: Thanks for your useful comments. It was an oversight on our part not to emphasize the characteristics of the AR-z spectrum. The AR-z spectrum is a detection method, which only is meant for signal detection, its amplitudes have only a relative meaning, not an absolute meaning; namely, if we find a spectral peak over the background noise level, it can be identified as the quasi-harmonic signal, while the absolute amplitude of the AR-z spectral peak contains no direct information about the actual complex amplitude of the detected signal.

Here we first give an example of the normal mode ${}_0S_2$ of the free oscillation, as shown in Figure R6. We can see that the real excited amplitudes for the $m = \pm 1$ singlets are less than those of $m = \pm 2$ singlets from the Fourier spectrum, while in the AR-z spectrum, the 'amplitudes' of the $m = \pm 1$ singlets are even larger than those of $m = \pm 2$ singlets; and the SNRs and frequency resolution are highly improved than the Fourier spectrum.

Figure R6. The normalized Fourier and AR-z spectra of 0S_2 from the Moxa SG station after the 2004 Sumatra earthquake.

As for the AR-z spectrum for the PM (as shown in Figure R7), the red area denotes the background noise level (the mean amplitudes which were calculated after removing some strong signals), we can see that the signals indicated in the main text are all over this background level, and can be considered as signals. According to the above, the absolute ‘amplitudes’ of those spectral peaks are not indications of the absolute signals’ power. This result can indicate that the 7.3yr and 5.9yr signals have both positive and negative components. Even if we don’t consider the positive component of the 7.3yr signal, it still cannot be the ICW. As a prograde motion, the ICW can only be present in the positive part, namely, it has no negative component, which is similar to the well-known Chandler wobble.

Figure R7. The AR-z spectrum of the PM.

Based on your comments, we added some notes, please see lines 72-77: “Different from the oceanic tidal signals that have both prograde and retrograde components in PM²²⁻²⁴, the ICW is a prograde motion (the same as the Chandler wobble, i.e., the mantle wobble); in the complex spectra of the PM, a prograde/retrograde wobble only has a positive/negative frequency. The identification of the well-known Chandler wobble is based on this feature²⁵⁻²⁸. Therefore, as a prograde motion, the ICW only appears on the positive frequency axis of the PM spectrum and this is a distinguishing feature for identifying it.”, and lines 83-85: “(The AR-z method is meant for signal detection, the amplitude of it contains no direct information about the actual complex amplitude of the detected signal)”.

Let us accept that the 8.5 yr peak is the ICW. This is not the period predicted from previous theoretical models that account for Earth structure and match other observed wobbles and nutations (e.g., refs 1-6). Why does the period not match those models and predictions? This must tell us something about the structure of the inner core.

I grant that these are not the questions that the authors have focused on, but I think the existence of the other peaks and the mismatch between predicted and “observed” ICW periods are both points that are too significant to be left unexplored and unexplained.

Response: Thanks for your constructive comments. The 8.5yr signal does not match the predicted periods from previous theoretical models; this is similar to the well-known Chandler wobble, which has a ~430 days observed period but a ~400 days theoretical period. As we know, the density jump $\Delta\rho_{ICB}$ at the ICB is still not well constrained, different Earth model has different $\Delta\rho_{ICB}$ (such as the 1066A, PREM, AK135, CORE11 models), while the period of the ICW is directly dependent on the $\Delta\rho_{ICB}$. Therefore, we believe our result (8.5yr period) is mainly because of that the $\Delta\rho_{ICB}$ of the real Earth is different from those of the models, and hence can be used to constrain the $\Delta\rho_{ICB}$ at the ICB.

According to your comments, we added the related parts to explain what can we learn from the ‘observed’ period of the ICW. Please see lines 183-189: “Our observed ICW period is slightly larger than the theoretical values (6.6-7.8yr)^{1,5-7}, but considering that even the generally accepted Chandler wobble observation period of ~430 days is ~30 days longer than its theoretical periods and that the density jump $\Delta\rho_{ICB}$ at the ICB was also poorly determined, this deviation is accepted. Considering this

newly determined period of the ICW, we can also invert the density jump $\Delta\rho_{\text{ICB}}$. Taking the density profiles of the PREM model as a reference, we finally obtained $\Delta\rho_{\text{ICB}}= 0.520\pm 0.05 \text{ g/cm}^3$ (see Methods), which is smaller than that of the PREM model (0.598 g/cm^3).”; and lines 195-196: “Besides, the density jump of $0.520\pm 0.05 \text{ g/cm}^3$ at the ICB is also inverted based on the observed ICW period.”; and the ‘Method’ parts: “**Constraint for the density jump at ICB. ...**” (please see lines 239-255).

Moderate points

•L99: How is the time series for the rate of change of ΔLOD constructed? I don’t think it is provided directly by the data service; so, the authors have presumably calculated it. If so, what method was used and was the differencing done before/after filtering of ΔLOD ? Or did the authors calculate the derivative of their fit to the ΔLOD data? Such details would be useful for anyone trying to reproduce the work.

Response: Thanks for your comments. As for the $d\Delta\text{LOD}/dt$ (here we refer it as to $g(t)$), because the amplitudes of the signals contained in the ΔLOD may be time-varying, so it’s hard to select suitable functions to fit the ΔLOD data, and such fitting process may cause some deviations; hence, we directly use a classical discrete numerical derivation algorithm, i.e., $g(t_i)=[\Delta\text{LOD}(t_{i+1})-\Delta\text{LOD}(t_i)]/\Delta t$.

Given that the final used ΔLOD has a time span of 1900-2020, which was reconstructed by two different records with different sample intervals. The first ΔLOD record is a long-term dataset (1623/06-2008/06) with 1-year sampling from IERS; while the second record is the EOPC04 ΔLOD (1962/01-2019/12) with 1-day sampling. To merge those two records, we first need to resample them to the same interval, so we used a low-pass filter for the second record. If no filter was used, the high-frequency signals will be leaked to the low-frequency parts, i.e., the so-called aliasing signals. Here we show a comparison of this.

From Figure R8a, we can see that the down-sampled ΔLOD with and without a low-pass filter have clear differences, such differences are caused by the high-frequency signals’ leak. Figure R8b can more clearly show this, we can see that only with a low-pass filter, the obtained down-sampled ΔLOD has the same low-frequency parts as the original record. If we didn’t use a filter before the down-sampling process, the final obtained result for the fitted $\sim 8.5\text{yr}$ certainly will have some difference from our used recommended result (as Figure R9 shows). Figure R9 shows the results from those two different records have some amplitude differences ($d\Delta\text{LOD}/dt$ have similar results, we will not further

show them). Thus, we recommend the result obtained from the Δ LOD with a low-pass filter, as filtering before a down-sampling process is the standard process in preprocessing.

According to this comment and your following comments and suggestions, we added a new part in the 'Methods' to explain those related problems. Please see the following response.

Figure R8. (a) the original Δ LOD and the down-sampled Δ LOD (with and without using a low-pass filter). (b) the corresponding Fourier amplitude spectra of the records in (a).

Figure R9. The fitted 8.5yr oscillation from the Δ LOD with and without using a low-pass filter.

•Related to the above, it would be useful to have a specific data section in the Methods where the sources and full processing flows of each time series were clearly laid out rather than needing to pick the details out of the caption to supplementary figure 1.

Response: Thanks for your useful suggestions. We once want to explain those in the main text, but considering the length limitation, we removed them to the SI the last time. This time, according to your suggestions, we added a new part “**Datasets and Preprocessing**” in the Methods. Please see lines 257-274: “The PM observations were obtained from the EOPCO1 dataset (1861/01-1889/12 with 0.1yr sampling and 1900/01-2019/12 with 0.05yr sampling); the Δ LOD record was combined with a long-term dataset⁴⁵ (1623/06-2008/06 with 1yr sampling from IERS; EOPCO1) and the EOPCO4 Δ LOD record⁴⁶ (1962/01-2019/12 with 1-day sampling); the AAM (1949/01-2019/12, sampling at 6 hours) record was from the Special Bureau for the Atmosphere⁴⁷⁻⁴⁹. The AAM was calculated from NCEP/NCAR reanalyses archived on pressure surfaces, and the inverted barometer (IB) pressure term was chosen as the mass term. The OAM record was obtained from the Special Bureau for the Oceans' datasets: ECCO_50yr⁵⁰ (1949/01-2003/01, sampling at 10 days) and ECCO_kf080i⁵¹ (1993/01-2020/3, sampling at 1 day). To standardize the sampling intervals of the records, we down-sampled all records to 1yr, and to avoid aliasing effects in this down-sampling process, a low-pass filter (with a cut-off frequency $f_c=0.5$ cpy) was used prior to down-sampling.

Note that although the theoretical amplitudes of the 8.85yr and 9.3yr zonal tides are quite small (only $\sim 2 \mu\text{s}$ for the 9.3yr tide and less than $1 \mu\text{s}$ for the 8.85yr tide) and far less than the background noise level of the Δ LOD time series, they were removed from this Δ LOD record based on a given model⁵² to avoid the effects of some well-known signals on the target ~ 8 yr period band. The $d\Delta\text{LOD}/dt$ was obtained by a classical discrete numerical derivation algorithm, i.e., $d\Delta\text{LOD}(t_i)/dt=[\Delta\text{LOD}(t_{i+1})-\Delta\text{LOD}(t_i)]/\Delta t$.”.

•L108-110: I do not follow the argument here. What is “it” referring to in the phrase “...only when it is in the same plane...”? What do you mean by being in the same plane as the tilt angle? The axial torque vector Γ_z is always in the same plane as the vector defining the inner core figure axis, since any two (non-colinear) vectors from the origin define a plane.

Response: We are sorry for our lack of clarity here and thank you for catching this oversight. As you said, the two vectors the static tilted axis Ω'_m and the rotation axis of the mantle Ω_m will define a plane,

and the axial torque Γ_z will reach its peak/valley when Γ_z is in the plane. So, we restate it as: “The axial torque Γ_z reaches its peak/trough only when the Γ_z is in the plane defined by the static tilted axis Ω'_m and the rotation axis of the mantle Ω_m .” Please see lines 121-122.

•The Methods section shows how the equatorial and axial torques on the mantle are related to the changes in mantle rotation vector (observed polar motion and LOD signals); however, there is no explicit demonstration that the ratio of those torques equates to the tilt angle of the inner core. Given that the complete description of couple rotations and torques for the mantle-outer core-inner core system is far from trivial (particularly if we allow that the solid regions of the Earth may be somewhat triaxial), I think the authors should clearly articulate the reasoning here.

Response: Thanks for your valuable reminder. When the inner core wobbles around a tilted axis relative to the mantle, the torque of mantle-inner core gravitational coupling Γ_{MICG} in the equatorial plane of the inner core causes the angular momentum exchange between the mantle and the inner core. This torque Γ_{MICG} produces a component Γ_{eq} in the equatorial plane of the mantle, which changes the Earth's polar motion, and produces a component Γ_z in the direction of Earth's rotation, which is perpendicular to the equatorial plane of the mantle, and Γ_z causes the ΔLOD (length-of-day variation). Therefore, according to the vector addition, the angle between the inner core's equatorial plane and the mantle's equatorial plane is $\arctan(|\Gamma_z|/|\Gamma_{eq}|) = 0.17^\circ$, which is also the tilted angle of the inner core. To further illustrate this problem, we draw a simple schematic in Figure R9. We can see that the angle θ between the equatorial plane of the inner core and the equatorial plane of the mantle is $\arctan(|\Gamma_z|/|\Gamma_{eq}|)$.

Figure R9. The red circle represents the equatorial plane of the inner core, and the gray circle represents the equatorial plane of the mantle. The torque of mantle-inner core gravitational coupling Γ_{MICG} moves on the equatorial plane of the inner core because of the inner core wobble; the Γ_{MICG} can

be decomposed into the vector sum of the torque Γ_{eq} on the equatorial plane of the mantle and the torque Γ_z perpendicular to the equatorial plane of the mantle.

According to your comments, we added Figure R9 in the new Figure 4 of the main text and added a note in line 153.

Minor points

•L21: Repetition of “solid inner core” one of which should be “solid mantle”.

Response: Done as your reminder, thanks!

•L35: “and etc.” is unnecessary.

Response: Deleted it as your suggestion.

•L67: “that has a significantly different feature” Different than what?

Response: Thanks for your reminder! In the new MS, we modified this sentence as: “Different from the oceanic tidal signals that have both prograde and retrograde components in PM²²⁻²⁴, the ICW is a prograde motion (the same as the Chandler wobble, i.e., the mantle wobble); in the complex spectra of the PM, a prograde/retrograde wobble only has a positive/negative frequency. The identification of the well-known Chandler wobble is based on this feature²⁵⁻²⁸. Therefore, as a prograde motion, the ICW only appears on the positive frequency axis of the PM spectrum and this is a distinguishing feature for identifying it.”. Please see lines 72-77.

•L106: “using a more complicated method” is too vague. You don’t have to explain the method here but should at least be specific on what it was.

Response: Thanks for your suggestion. We modified it as: “the extracted oscillations using a more complicated method (the normal time-frequency transform, NTFT) show almost the same results (see Figure S3).”. Please see line 118 and 119.

•L108- I would use “trough” rather than “valley”.

Response: Done as your suggestion. Please see line 121.

•L192- Am and Cm should have units of kg m².

Response: Thanks for your kind reminder! We added the unit. Please see line **216**.

REVIEWER COMMENTS

Reviewer #1 (Remarks to the Author):

Thanks for the detailed and comprehensive responses, which are highly appreciated.

The manuscript has significantly improved relative to the original version, at least in its part of making connections with seismological findings. Another reviewer had similar concerns, which helped improve the links with the seismological work and corresponding discussions.

I now have minor concerns remaining that I hope can be addressed. However, they are optional, and it is between the Editor and you to consider if you want to implement them.

- I find the new abstract version better than the previous one, but still a bit unpolished. As far as I know, the use of wording such as “for the first time, we newly find...” or “Our novel finding...” is discouraged. I suggest removing it.

- I also suggest removing the statement “(which is also a pending problem)” at the end of the abstract. Like many other parameters, such as anisotropy, lateral heterogeneity, and differential rotation, the inner-outer core density contrast is a subject of seismological studies and comes with uncertainty. Your inferred value of 0.52 g.cm^3 (by the way, you should remove the last zero from the value reported in the abstract because your uncertainty estimate is 0.05) is a valuable contribution and stands on its own without explicitly stating that it is “a pending problem.” A relatively recent review of the inner-outer core density contrast and the differential rotation, from a seismological point of view, is given in the inner core book. You can consider adding this reference to the seismological references made at the end of the first paragraph of the introduction: <https://www.cambridge.org/core/books/earths-inner-core/3994E0608D352380FAF75165A4125DDD>

- Line 135-136: I suggest removing “although it has some uncertainties,” as from your referencing, it looks as though the first study contradicts the second, although they were published parallelly supporting the same model.

- Line 138: there is a problem here as you cite two studies (#36 and #37) that found hemispherical differences in a) velocity and b) attenuation. Note that this is not the same as a hemispherical difference in anisotropy. In fact, such a dynamical model (proposed by #34 and #35) would erase any anisotropic

signature, which is why some contest it. Following this, I have difficulties understanding how referencing #39 can follow from this discussion. If the proposed geodynamical model that would create a hemispherical difference (#34 and #35) is indeed in operation, it will contradict anisotropy at the top of the inner core. I think you are safer excluding anisotropy completely from your discussions where you try to establish a link with seismological observations. Many readers will not understand why anisotropy is brought into these discussions in the first place.

- I don't understand the last sentence of the main part of the manuscript (before the Methods). You are saying that the static tilt should be used as a constraint on the inner core structure retrieved from seismology, but it is unclear in what sense. For example, if it is meant that it should be used in the sense that a tilt would also require a density jump of 0.52 g/cm^3 at the ICB or that it would require a differential rotation much less than 1.0 degrees per year, then OK. But if it is meant in the sense that it would require some sort of a travel time correction, then this wouldn't make much sense because 0.17 degrees is less than 20 km on the Earth's surface and negligible at the ICB in comparison with the wavelengths the seismic probes are sensitive to. At the Earth's surface, this approaches the error in the hypocenter location, and it is within the uncertainty we deal with in seismology. It would be good to clarify what you meant.

Reviewer #2 (Remarks to the Author):

The authors have answered properly my concerns and met all my suggestions. Besides, I think the manuscript has been improved with the comments of the other reviewers. Therefore, I am satisfied with the revision and can recommend it for publication.

There are some minor changes I'd like to suggest. They are:

L 15 "This tilt implies that..." sounds to a proven causality relation. I'd suggest changing it to "this tilt is consistent with the assumption that...". Or rewording the sentence with the same idea to make it more compact

L 21 "The solid Earth consists mainly of...". As the result is based on Earth rotation data, and quite a lot of people might not be familiar with the current theory of earth rotation, I'd suggest modifying the first sentence so that it recalls somehow the results are based on earth rotation theory. For instance, you might start this way: "Current theories of the Earth rotation consider a basic model of the solid Earth that consists mainly of..."

L 22-23 ff. The paper has many references to the “axes of figure” of earth components. However, I’m afraid that most readers won’t be familiar with the concept of axis of figure used in earth rotation, even though many of them might work in that field. My guess is because the exact definition of this concept is not simple at all but takes up more than 4 pages (with 34 Eqs.!) of Appendix A in the reference 2, i.e., the 1991 paper by Mathews et al - in which the so-called three-layer earth model was introduced as well as the associated four normal rotational modes (2 known + 2 new ones due to the SIC). Of course, it will be out of place providing details of such intricate definitions, but it may be appropriate to recall that the concept of axis of figure used in a work relying on earth rotation data is (and must be) the one used in the corresponding earth rotation theory and resulting equations needed to obtain those data. I think this can be made straightforward, e.g., by including a footnote (or a comment) at the first appearance of the word “axis”, clarifying that figure axis is ever understood as defined in appendix A of the current ref. 2.

L 25 Please remove “geometric”, it may be confusing

L 55 ...they were... -> ... they are...

L 84 I think the explanation might be slightly more detailed. For instance: “for signal detection” -> “for the accurate detection of the frequencies of harmonic components of a signal”

L 119 NTFT needs a reference

L 169 - Eq (1) - The notation for the equatorial torque exerted for the mantle on the SIC used in the left-hand side of the equation is the same as that used in the right-hand side of Eq (6) in L 214, where it refers to the total torque. Furthermore, χ and m in (6) refer to the observed (total) excitation and polar motion, whereas the χ/m used in L 170-172 correspond only to the 8.5-year component, as implicit in the calculations described in L 171-172. This may raise some confusion that could even lead to a serious misunderstanding. Therefore, I suggest either changing the notation or clarifying the situation by explaining that the same notation is being used for ease of visualization but that the torques, complex PM and excitation functions in Eqs 1 & 2 correspond only to the determined 8.5 yr signal as opposed to the same symbols when they appear in Eqs 6 & 7. Perhaps the second option is clearer; you could also combine both clarifications.

L 183-186. Concerning the values of the periods of rotational normal modes derived either from geophysical earth models or from earth rotation data, FCN (free core nutation) has been observed over 30 years and its period is also a good example of how much those values can differ. A good sample of values is available in table 3 of (doi: 10.1093/gji/ggab079); the rule there seems to be that ‘theoretic’

and 'observed' periods never match each other. Please consider whether you wish to include a comment on FCN too.

L 210-211 I think you should explain in the text, here or earlier, that the mantle inertia ellipsoid is assumed to have axial symmetry. I know this is the usual assumption in most of Earth rotation related work, but NatCom readers are a much wider audience.

L 219-220 Some readers may not be very familiar with the various estimates of the CW complex period, particularly those of its Q factor, which exhibit greater variability. I suggest including a sample of some reference values/intervals and the corresponding bibliography. You could include, for example, a reference to a table from a recent and easy to find article. That way it will be evident that the complex factor that appear in this line is very close to 1.

L 223 Combing -> Combining

L 258 EOPCO1 -> EOPC01

Reviewer #3 (Remarks to the Author):

I appreciate the efforts made by the authors in response to the previous round of reviews. The work is improved; however, I still have some concerns.

Major points

The translation from torques to tilt

It remains unclear to me why the torques deduced for the equatorial and axial components resolve the inner core tilt. Certainly, the two vectors can be combined in this way, but I see no clear demonstration that the torques on the mantle that cause polar motion and LOD variations should lie in the equator of the inner core aligned with the longitude of the tilt. LOD changes arising from inner core-mantle gravitational coupling have often been attributed to a spherical harmonic degree 2, order 2 component of the inner core shape (e.g. Dumberry and Manda, 2021; Buffett, 1996), in which case the axial torque has no relation to the tilted figure axis. That mechanism would imply a triaxial inner core, but even if the authors wish to consider the case of an oblate spheroidal inner core, then it seems to me that a gravitational attraction between a tilted inner core and the mantle would result in an equatorial torque aligned with the intersection of the inner core and mantle equatorial planes. Perhaps I am thinking

about the dynamics incorrectly, but since this calculation is central to the argument for the tilt, I think that the theory needs to be more fully explained.

The framing of previous geodynamic work suggesting a large inner-core tilt

While it is true that Greiner-Mai et al (2000, ref. 8 of the manuscript) propose alignment of the inner core spin axis with the geomagnetic dipole axis as a hypothesis from which to start their analysis, I do not think that this is a well-founded and widely accepted idea. Greiner-Mai & Barthelmes (2001) argue from polar motions data for an inner-core figure axis tilted by about 1 degree relative to the mantle rotation axis. Dumberry and Manda (2021) argue from gravity data for constraints on the inner-core figure axis tilt on the order of 0.014 - 0.07 degrees. Just as I do not think that the weight of previous seismic studies supports a large inner core tilt, I do not think that the overall view of previous geodynamic studies favours a large tilt.

Moderate points

- L25-26, L31-32, L121-123 and figure 4: From the outset I think that the potential offset between the inner core rotation axis and the mantle rotation axis should be mentioned. L25-26 imply that they should be the same, but the main point of the paper is to argue that they are not. Therefore, the first mention of the inner core and mantle rotation axes should make that distinction and use the Ω_m and Ω_m' notation used elsewhere in the manuscript.
- I had not appreciated that the amplitude information from the AR-z process was not particularly meaningful, thank you for that clarification. I think that the background noise level indication in the figure included in the response should also be used in the paper (figure 2a) to illustrate whether the various peaks rise above the detection threshold. I remain concerned that the other peaks detected in the PM and LOD time series have no apparent physical explanation, but the reasoning for choosing that one peak as indicative of the ICW is now clearer.

Minor points

- L14: “than that”  “than the”
- L41: a citation to Triana et al (2022) might be included as another recent review relevant to the question of the importance of the topic.
- L89-90: “Since there is no other mechanism has been proposed...”  “Since no other mechanism has been proposed...”
- L95: “super frequency resolution”, the use of “super” here feels awkward to me.
- L117: “ICWs”  “ICW signals” or similar
- L167: “inner-mantle” is missing “core”

References

Buffett, B. A. Gravitational oscillations in the length of day. *Geophys Res Lett* 23, 2279–2282 (1996).

Dumberry, M. & Manda, M. Gravity Variations and Ground Deformations Resulting from Core Dynamics. *Surv Geophys* 1–35 (2021) doi:10.1007/s10712-021-09656-2.

Greiner-mai, H. & Barthelmes, F. Relative wobble of the Earth's inner core derived from polar motion and associated gravity variations. *Geophys J Int* 144, 27–36 (2001).

Triana, S. A. et al. Core Eigenmodes and their Impact on the Earth's Rotation. *Surv Geophys* 43, 107–148 (2022).

Response to Reviewers

#####

#####

Reviewer #1 (Remarks to the Author):

Thanks for the detailed and comprehensive responses, which are highly appreciated. The manuscript has significantly improved relative to the original version, at least in its part of making connections with seismological findings. Another reviewer had similar concerns, which helped improve the links with the seismological work and corresponding discussions.

Response: We appreciate this positive assessment and would like to thank you for the thoughtful review and the insightful comments to help improve this study. We have carefully considered your suggestion and revised this MS according to the comments, and hope that the revised version is more suitable.

I now have minor concerns remaining that I hope can be addressed. However, they are optional, and it is between the Editor and you to consider if you want to implement them.

- I find the new abstract version better than the previous one, but still a bit unpolished. As far as I know, the use of wording such as “for the first time, we newly find...” or “Our novel finding...” is discouraged. I suggest removing it.

Response: Thank you very much for your kind reminder, we have changed the related expressions in the new MS, please see lines 11 and 13.

- I also suggest removing the statement “(which is also a pending problem)” at the end of the abstract. Like many other parameters, such as anisotropy, lateral heterogeneity, and differential rotation, the inner-outer core density contrast is a subject of seismological studies and comes with uncertainty. Your inferred value of 0.52 g.cm³ (by the way, you should remove the last zero from the value reported in the abstract because your uncertainty estimate is 0.05) is a valuable contribution and stands on its own without explicitly stating that it is “a pending problem.” A relatively recent review of the inner-outer core density contract and the differential rotation, from a seismological point of view, is given in the inner core book. You can consider adding this reference to the seismological references made at the

end of the first paragraph of the introduction: <https://www.cambridge.org/core/books/earths-inner-core/3994E0608D352380FAF75165A4125DDD>

Response: Thank you for your useful suggestions and for bringing a useful textbook to our attention, which greatly helps me to track the progress of seismological observations of the inner core and deepen my understanding of the structure of the inner core. We fully agree with your comments about the inner-outer core density contrast, and we have removed the latest sentence at the end of the abstract (please see line 17) and deleted the last zero in “0.520 g/cm³” (please see lines 17, 209, and 216), and we also added a reference as you suggested (please see lines 34, 53, and 207).

- Line 135-136: I suggest removing “although it has some uncertainties,” as from your referencing, it looks as though the first study contradicts the second, although they were published parallelly supporting the same model.

Response: Thank you for your kind reminder, we have removed it in the new MS, please see line 155.

- Line 138: there is a problem here as you cite two studies (#36 and #37) that found hemispherical differences in a) velocity and b) attenuation. Note that this is not the same as a hemispherical difference in anisotropy. In fact, such a dynamical model (proposed by #34 and #35) would erase any anisotropic signature, which is why some contest it. Following this, I have difficulties understanding how referencing #39 can follow from this discussion. If the proposed geodynamical model that would create a hemispherical difference (#34 and #35) is indeed in operation, it will contradict anisotropy at the top of the inner core. I think you are safer excluding anisotropy completely from your discussions where you try to establish a link with seismological observations. Many readers will not understand why anisotropy is brought into these discussions in the first place.

Response: Thank you for your suggestion, we found some discussions and statements in the old MS are really confusing, so we have deleted and revised those statements. We prefer to retain some presentations of the anisotropy, as the results of the studies may be relevant to the further conclusions we suggest for the northwestern hemisphere. The specific modification is expressed as follows: “To explain the asymmetry between the inner core's eastern and western hemispheres in seismological observations^{42,43}, a previous dynamical model considers the crystallization and melting at the surface of the inner core and has similar suggestions⁴⁴, i.e., the western hemisphere of the inner core is denser

than its eastern hemisphere. Interestingly, a seismological study has suggested that the western hemisphere of the inner core may be relatively denser⁴⁵, and a thicker compacting layer at the top of the inner core's western hemisphere was also suggested⁴⁶; a more nuanced research found that the western zone is largely confined to the northern hemisphere⁴⁷, please see lines 151-158.

- I don't understand the last sentence of the main part of the manuscript (before the Methods). You are saying that the static tilt should be used as a constraint on the inner core structure retrieved from seismology, but it is unclear in what sense. For example, if it is meant that it should be used in the sense that a tilt would also require a density jump of 0.52 g/cm^3 at the ICB or that it would require a differential rotation much less than 1.0 degrees per year, then OK. But if it is meant in the sense that it would require some sort of a travel time correction, then this wouldn't make much sense because 0.17 degrees is less than 20 km on the Earth's surface and negligible at the ICB in comparison with the wavelengths the seismic probes are sensitive to. At the Earth's surface, this approaches the error in the hypocenter location, and it is within the uncertainty we deal with in seismology. It would be good to clarify what you meant.

Response: Thanks for your constructive suggestions! This sentence is really vague. We originally intended to say that this static tilt could *qualitatively* constrain some core structures or motions from seismic observations, such as the density differences between the eastern and western hemispheres, the inner core differential rotation, or inner core oscillations. As you commented, it is currently very difficult to quantitatively constrain the inner structures obtained from seismic waves. We have therefore revised this sentence to: and we suggest such consistency should be helpful to the inner core oscillation or differential rotation (please see lines 220-222).

#####

Reviewer #2 (Remarks to the Author):

The authors have answered properly my concerns and met all my suggestions. Besides, I think the manuscript has been improved with the comments of the other reviewers. Therefore, I am satisfied with the revision and can recommend it for publication.

Response: Thank you very much for your thorough read of our MS and insightful comments. We carefully addressed each of your comments and updated the manuscript accordingly. Please see our detailed responses below with tracking for the exact edits.

There are some minor changes I'd like to suggest. They are:

L 15 "This tilt implies that..." sounds to a proven causality relation. I'd suggest changing it to "this tilt is consistent with the assumption that...". Or rewording the sentence with the same idea to make it more compact

Response: Thank you for the suggestions to make it more readable, we changed it as you suggested, please see line 15.

L 21 "The solid Earth consists mainly of...". As the result is based on Earth rotation data, and quite a lot people might not be familiar with the current theory of earth rotation, I'd suggest modifying the first sentence so that it recalls somehow the results are based on earth rotation theory. For instance, you might start this way: "Current theories of the Earth rotation consider a basic model of the solid Earth that consists mainly of..."

Response: Thanks very much for your suggestions. According to your and Reviewer #3's suggestions, we have made detailed revisions here, adding the background on the theories of the Earth's rotation. Please see lines 21-27.

L 22-23 ff. The paper has many references to the "axes of figure" of earth components. However, I'm afraid that most readers won't be familiar with the concept of axis of figure used in earth rotation, even though many of them might work in that field. My guess is because the exact definition of this concept is not simple at all but takes up more than 4 pages (with 34 Eqs.!) of Appendix A in the reference 2, i.e., the 1991 paper by Mathews et al - in which the so-called three-layer earth model was introduced as well as the associated four normal rotational modes (2 known + 2 new ones due to the SIC). Of course, it will be out of place providing details of such intricate definitions, but it may be appropriate to recall that the concept of axis of figure used in a work relying on earth rotation data is (and must be) the one used in the corresponding earth rotation theory and resulting equations needed to obtain those data. I think this can be made straightforward, e.g., by including a footnote (or a comment) at the first

appearance of the word “axis”, clarifying that figure axis is ever understood as defined in appendix A of the current ref. 2.

Response: Thanks a lot for your very useful suggestions, we have added a note about this according to your suggestions, please see line 26.

L 25 Please remove “geometric”, it may be confusing

Response: Done as you suggested. Please see line 25.

L 55 ...they were... -> ... they are...

Response: Done as you suggested. Please see line 67.

L 84 I think the explanation might be slightly more detailed. For instance: “for signal detection” -> “for the accurate detection of the frequencies of harmonic components of a signal”

Response: Done as you suggested. Please see line 96.

L 119 NTFT needs a reference

Response: Done as you suggested. Please see line 137.

L 169 - Eq (1) - The notation for the equatorial torque exerted for the mantle on the SIC used in the left-hand side of the equation is the same as that used in the right-hand side of Eq (6) in L 214, where it refers to the total torque. Furthermore, χ and m in (6) refer to the observed (total) excitation and polar motion, whereas the χ/m used in L 170-172 correspond only to the 8.5-year component, as implicit in the calculations described in L 171-172. This may raise some confusion that could even lead to a serious misunderstanding. Therefore, I suggest either changing the notation or clarifying the situation by explaining that the same notation is being used for ease of visualization but that the torques, complex PM and excitation functions in Eqs 1 & 2 correspond only to the determined 8.5 yr signal as opposed to the same symbols when they appear in Eqs 6 & 7. Perhaps the second option is clearer; you could also combine both clarifications.

Response: Thank you for the suggestion to make it more readable accessible. We have changed it as your suggestions, please see lines 176 and 187-202.

L 183-186. Concerning the values of the periods of rotational normal modes derived either from geophysical earth models or from earth rotation data, FCN (free core nutation) has been observed over 30 years and its period is also a good example of how much those values can differ. A good sample of values is available in table 3 of (doi: 10.1093/gji/ggab079); the rule there seems to be that ‘theoretic’ and ‘observed’ periods never match each other. Please consider whether you wish to include a comment on FCN too.

Response: This is a good idea, we have added a comment on FCN “free core nutation observation period of retrograde ~430 days is ~20 days shorter than its theoretical periods^{49,50}” as you suggested, please see lines 205 and 206.

L 210-211 I think you should explain in the text, here or earlier, that the mantle inertia ellipsoid is assumed to have axial symmetry. I know this is the usual assumption in most of Earth rotation related work, but NatCom readers are a much wider audience.

Response: This is a good idea, we have made this change as your suggestions, please see lines 230 and 231.

L 219-220 Some readers may not be very familiar with the various estimates of the CW complex period, particularly those of its Q factor, which exhibit greater variability. I suggest including a sample of some reference values/intervals and the corresponding bibliography. You could include, for example, a reference to a table from a recent and easy to find article. That way it will be evident that the complex factor that appear in this line is very close to 1.

Response: Thank you for your insightful comments and suggestions, we have made corresponding changes and added relevant references as you suggested. Please see lines 241 and 242.

L 223 Combing -> Combining

Response: Our apologies for this oversight. We modified it according to your comment. Please see line 246.

L 258 EOPCO1 -> EOPC01

Response: Thanks for your kind reminder! We modified it as your comment. Please see line 281.

Sincerely

Jose M Ferrandiz

#####

Reviewer #3 (Remarks to the Author):

I appreciate the efforts made by the authors in response to the previous round of reviews. The work is improved; however, I still have some concerns.

Response: Thank you very much for some recognition of our work in the last round and thoughtful review. The remaining concerns and the helpful comments drive us to further improve this MS. We carefully addressed each of your comments and modified this MS based on your helpful feedback. Please see our detailed responses below with tracking for the exact edits.

Major points

The translation from torques to tilt

It remains unclear to me why the torques deduced for the equatorial and axial components resolve the inner core tilt. Certainly, the two vectors can be combined in this way, but I see no clear demonstration that the torques on the mantle that cause polar motion and LOD variations should lie in the equator of the inner core aligned with the longitude of the tilt. LOD changes arising from inner core-mantle gravitational coupling have often been attributed to a spherical harmonic degree 2, order 2 component of the inner core shape (e.g. Dumberry and Mandeau, 2021; Buffett, 1996), in which case the axial torque has no relation to the tilted figure axis. That mechanism would imply a triaxial inner core, but even if the authors wish to consider the case of an oblate spheroidal inner core, then it seems to me that a gravitational attraction between a tilted inner core and the mantle would result in an equatorial torque aligned with the intersection of the inner core and mantle equatorial planes. Perhaps I am thinking about the dynamics incorrectly, but since this calculation is central to the argument for the tilt, I think that the theory needs to be more fully explained.

Response: Thank you for your insightful comments, in an ideal Earth model that the symmetry axes

of mantle's elliptical surfaces of constant density are aligned along the axis of rotation (see Figure 10a), some cases are indeed as you commented. Here, we consider a more realistic and complex Earth model that the symmetry axes of elliptical surfaces of constant density are no longer aligned for homogeneous mantle (see Figure 10b), but this was not specifically stated in the previous drafts. Next, we will explain in detail why the torques deduced for the equatorial and axial components resolve the inner core tilt. We first approach the problem of the mantle-inner core gravitational (MICG) torque. The torque that changes the polar motion (PM) is always on the equatorial plane of the mantle, and the torque that changes the ΔLOD is always aligned with the rotation axis of the mantle. In other words, any torque exerted on the equatorial plane of the mantle can change the PM, and any torque exerted on the rotation axis of the mantle can change the ΔLOD . The $\sim 8.5\text{yr}$ ICW signal is detected from the PM, demonstrating that there is a torque Γ_{eq} always in the equatorial plane of the mantle (the gray plane in Figure 4) which changes its direction with the period of $\sim 8.5\text{yr}$. However, the $\sim 8.5\text{yr}$ ICW signal is also detected in the ΔLOD , which demonstrates that there should be a torque Γ_z in the direction of the mantle's rotation which changes its magnitude with the period of $\sim 8.5\text{yr}$. Given that the interaction between the mantle and the inner core is mainly through the MICG coupling (as suggested by Buffett (1996) and Dumberry (2009)), the MICG torque can be targeted immediately and the two torques should be originated from the MICG torque (and hence can be further re-written as $\Gamma_{\text{eq}}^{\text{MICG}}$ and Γ_z^{MICG}). The only way these two torques $\Gamma_{\text{eq}}^{\text{MICG}}$ and Γ_z^{MICG} can be combined is by vector addition, creating the torque Γ_{MICG} which changes direction continuously in the tilted plane (blue plane in Figure 4) relative to the mantle. Next, we explain why the MICG torque Γ_{MICG} caused by the ICW is in the tilted plane and it is the mean equatorial plane of the inner core. Before that, we introduce the background of the Earth's rotation and the Earth model used. The mantle's rotation axis and its figure axis are very close due to the centrifugal force, and they can be seen as the same axis Ω_m . Similarly, the figure axis Ω_{ic} of the inner core is very close to its rotation axis Ω'_m (but much larger than the mantle) **which locates the direction of the lowest MICG potential energy** (equilibrium state; similar to the lowest point of a simple pendulum model). When the inner core's figure axis Ω_{ic} deviates from its rotation axis Ω'_m (the equilibrium state), it will excite a rotation mode that the inner core's figure axis Ω_{ic} wobbles around its rotation axis Ω'_m under the influence of spin, i.e., the ICW. Current theories of the Earth's rotation are based on the classical 2-D Earth model; in this model, the flat solid Earth

consists mainly of a solid inner core, a liquid outer core, and a solid mantle with the same center of mass; this model is reduced to a series of layered elliptical surfaces on which the density is constant, and the mantle's elliptical surfaces of constant density is aligned in the rotation's axis Ω_m (see Figure. R10a). Thus, the direction of the lowest MICG potential energy is the mantle's rotation axis Ω_m , and **the inner core's rotation axis Ω'_m is aligned with of the axis Ω_m** . However, for the real Earth, the surfaces of constant density of the highly heterogeneous mantle are no longer the ideal ellipsoids, and their symmetry axes may be slightly tilted randomly relative to the mantle's rotation axis Ω_m , especially with large uncertainties at the core-mantle boundary (CMB), as shown in Figure. R10b. The direction of the mantle's figure axis is only sensitive to the heterogeneity or ellipticity of the upper mantle according to its definition (the direction corresponding to the maximum moment of inertia). However, the closer matters have the stronger gravitational force, and therefore the lowest MICG potential energy state (the equilibrium state) is that the inner core's figure axis Ω_{ic} is aligned with the mean axis of the lower mantle's elliptical surfaces of constant density; the mean axis is usually tilted relative to the mantle's rotation axis Ω_m and it coincides with the inner core's rotation axis Ω'_m (which represents the direction of the lowest MICG potential energy). When the inner core's figure axis Ω_{ic} deviates from this equilibrium state, it excites the ICW that the figure axis Ω_{ic} wobbles around its rotation axis Ω'_m . In the process, the MICG torque Γ_{MICG} always brings the inner core back to the equilibrium state, that is, the torque Γ_{MICG} always perpendicular to Ω'_m , and therefore torque Γ_{MICG} always in the plane perpendicular to Ω'_m , which is the mean equatorial plane of the inner core (blue plane in Figure 4). The decomposition of the MICG torque at a certain moment is shown in Figure 4.

Privately, we have communicated with Prof. Benjamin F. Chao about this part for the heterogeneous mantle and have received positive supports (we are planning to work together to further develop the theory of Earth rotation in response to this problem). We also sent the MS to Prof. Chao, and he strongly suggested that we should include a description of the heterogeneous mantle and the alignment of the inner core with the mean axis of the lower mantle at the static equilibrium due to the stronger attraction of nearby objects in this MS, which may improve the theory of the Earth's rotation. To clarify, we added the background of the Earth's rotation 'The flat solid Earth consists mainly of a solid inner core, a liquid outer core, and a solid mantle with the same center of mass, which is reduced to a series of layered elliptical surfaces on which the density is constant in the classical model^{1,2}. The

current theories regarding the Earth's rotation involve the consideration of the mantle's elliptical surfaces of constant density, whose symmetry axes are aligned in the direction of rotation, and hydrostatic effects necessitate that the inner core's figure axis Ω_{ic} (as defined in Appendix A of ref. 2) and rotation axis Ω'_m are aligned with the mantle's figure or rotation axis Ω_m (which are nearly identical due to centrifugal torque) in order to maintain equilibrium. The presence of random torques acting on the inner core results in a slight tilt and further excites a prograde rotation mode known as the inner core wobble (ICW), i.e., the inner core's figure axis Ω_{ic} wobbles about its rotation axis Ω'_m ²⁻⁴ (also represents the direction of the lowest gravitational potential energy of the mantle-inner core system). The above 'tilt' between the Ω_{ic} and Ω'_m is a generally *dynamic* tilt' (please see lines 21-32), the description of a possible highly heterogeneous mantle for the real Earth 'However, the elliptical surfaces of constant density within the heterogeneous mantle may exhibit random tilting around its rotation axis Ω_m considering its solid properties, particularly with significant uncertainties at the core-mantle boundary (CMB). Consequently, the inner core's rotation axis Ω'_m , which signifies the direction of the static equilibrium of the inner core (and corresponds to the lowest gravitational potential energy of the mantle-inner core system¹⁰), was previously believed to deviate from alignment with the mantle's axis Ω_m and instead possess a tilt relative to it, and thereby the 'tilt' between the Ω'_m and Ω_m is called as *static* tilt.' (please see lines 35-42), and explanation of the torque Γ_{MICG} always in the mean equatorial plane of the inner core 'Next, we estimate the tilt θ . The heterogeneous mantle forms a tilted rotation axis of the inner core Ω'_m which has the lowest mantle-inner core gravitational (MICG) potential energy (the axis at the equilibrium state); the deviation of the inner core's figure axis Ω_{ic} from the equilibrium state caused by the ICW will result in a MICG restoring torque Γ_{MICG} which always brings the inner core back to the equilibrium state, that is, the torque Γ_{MICG} always in the plane (blue plane in Figure 4) perpendicular to the inner core's rotation axis Ω'_m ; the plane is also the mean equatorial plane of the inner core.' (please see lines 177-183). The mean equatorial plane of the inner core perpendicular to the inner core's rotation axis Ω'_m , so we have also further modified the Figure. 4 for easier understanding, please see line 171.

Figure R10. Schematic diagram of the mantle's elliptical surfaces of constant density. (a) The mantle in the classical 2-D Earth model which is used in current theories of the Earth's rotation; the all symmetry axes of the mantle's elliptical surfaces of constant density are aligned in the rotation's direction. (b) The more realistic model for the highly heterogeneous mantle; the symmetry axes of the mantle's elliptical surfaces of constant density are slightly tilted randomly relative to the mantle's rotation axis Ω_m (the tilt angle is exaggerated to show).

Secondly, we fully agree with you that the inner core (gravitational) oscillation mode can change the LOD and the axial torque caused by it has no relation to the tilted figure axis. Based on the rotationally symmetric (biaxiality) three-layered Earth, the classical Earth rotation theories predicted two rotation modes related to the inner core, the free inner core nutation (FICN; irrelevant to this study), inner core wobble (ICW; Mathews et al., 1991, 2002; Dumberry, 2009; Rochester et al., 2014). Considering the triaxiality of Earth, Buffett (1996) predicted a new inner core's rotation mode, the inner core (gravitational) oscillation, to explained the intradecadal fluctuation in the Δ LOD. The ICW and inner core oscillation are two completely different modes, and they have different theoretical periods. The ICW refers to that the inner core's figure axis wobbles counterclockwise (as seen from above the North Pole) around the mantle's axis in a circular motion, taking one cycle to complete; while inner core oscillation refers to that the inner core's rotation rate changes periodically relative to the mantle's rotation rate. The geophysical mechanisms of the ICW and inner oscillation are different;

the ICW is caused by the Earth's rotation effect and the equatorial torque Γ_{eq} which is caused by the misalignment of the degree-2 order-0 component of the inner core and mantle's shape; while the inner core oscillation is caused by the axial torque (Γ_z) which is from the misalignment of the degree-2 order-2 component of the inner core and mantle's shape. The two modes are decoupled from each other, they cause different results and appear in different observations due to different orders. Normally, the ICW mode only changes the polar motion with its period, and the inner core oscillation mode can cause the ΔLOD . Unless the inner core and mantle have the static tilt described above, the ICW does not appear in the ΔLOD .

Reference:

- Buffett, B. A. (1996). Gravitational oscillations in the length of day. *Geophysical Research Letters*, 23(17), 2279-2282.
- Dumberry, M. (2009). Influence of elastic deformations on the inner core wobble. *Geophysical Journal International*, 178(1), 57-64.
- Mathews, P. M., Buffett, B. A., Herring, T. A., & Shapiro, I. I. (1991). Forced nutations of the Earth: Influence of inner core dynamics: 1. Theory. *Journal of Geophysical Research: Solid Earth*, 96(B5), 8219-8242.
- Mathews, P. M., Herring, T. A., & Buffett, B. A. (2002). Modeling of nutation and precession: New nutation series for nonrigid Earth and insights into the Earth's interior. *Journal of Geophysical Research: Solid Earth*, 107(B4), ETG-3.
- Rochester, M. G., Crossley, D. J., & Zhang, Y. L. (2014). A new description of Earth's wobble modes using Clairaut coordinates: 1. Theory. *Geophysical Journal International*, 198(3), 1848-1877.

The framing of previous geodynamic work suggesting a large inner-core tilt

While it is true that Greiner-Mai et al (2000, ref. 8 of the manuscript) propose alignment of the inner core spin axis with the geomagnetic dipole axis as a hypothesis from which to start their analysis, I do not think that this is a well-founded and widely accepted idea. Greiner-Mai & Barthelmes (2001) argue from polar motions data for an inner-core figure axis tilted by about 1 degree relative to the mantle rotation axis. Dumberry and Mandea (2021) argue from gravity data for constraints on the inner-core figure axis tilt on the order of 0.014 - 0.07 degrees. Just as I do not think that the weight of previous

seismic studies supports a large inner core tilt, I do not think that the overall view of previous geodynamic studies favours a large tilt.

Response: Thanks for your useful comments. Greiner-Mai et al. (2000) assumed alignment of the inner core's rotation axis with the geomagnetic dipole axis and proposed a tilt of 10° of the inner core relative to the mantle, this 'tilt' actually refers to static tilt (deviation of Ω_m and Ω'_m). Subsequently, Greiner-Mai and Barthelmes (2001) calculated a tilt of 1° from the excitation sequence of the polar motion, this 'tilt' actually refers to dynamic tilt (deviation of Ω_{ic} and Ω'_m). These two concepts are different, based on your comments, and we distinguish them (please see lines 31-32 and 41-42) and explain the reason why they are produced (please see lines 23-31 and 35-41).

In fact, the inverted the dynamic tilt in Greiner-Mai and Barthelmes (2001) is from the entire frequency band of the excitation sequence of the polar motion, which contains not only the ICW signal, but also what they found $\sim 70\text{yr}$, $\sim 30\text{yr}$, $\sim 20\text{yr}$ and trend term with large amplitudes, so the result is relatively large. At present, a very small inner core's dynamic tilt (no more than 0.1°) has been agreed in the geodynamics (Dehant, et al., 1993; Dumberry and Bloxham, 2002; Guo et al., 2005; Dumberry, 2008; Ding et al., 2019), which also includes the dynamic tilt of 0.014° - 0.07° calculated by Dumberry and Manda (2021). In this study, the dynamic tilt angle of the inner core can be easily estimated from the amplitude and period of the ICW observed in the polar motion, i.e., we can infer the true amplitude of the ICW from the amplitude observed in the polar motion $\mathbf{m}_s = 3.685 \mathbf{m} = 3.685 \times 4.7 \text{ mas} = \sim 17.3 \text{ mas}$ (Dumberry, 2010), and hence the dynamic tilt is $\mathbf{n}_s = \mathbf{m}_s / \sigma_{\text{ICW}} = \sim 53673 \text{ mas} = \sim 0.015^\circ$ (Mathews et al., 1992; Dumberry, 2010), where the $\sigma_{\text{ICW}} = 1/(8.5 \times 365)$ is in unit of cycle per days (cpd); this dynamic tilt is consistent with the results of Dumberry and Manda (2021) and previous studies (Dehant, et al., 1993; Dumberry and Bloxham, 2002; Guo et al., 2005; Dumberry, 2008; Ding et al., 2019). The static tilt is not included in the current Earth models, and the current theories of Earth's rotation (Mathews et al., 1991, 2002; Dumberry, 2009; Rochester et al., 2014) are based on the existing Earth model, so the static tilt is not considered in theories of Earth's rotation. Although the static tilt postulated by Greiner-Mai et al. (2000) was large and unproven, their proposal represents a possibility. This is why we explain it that way in old MS. According to your comments, we have adjusted the relevant statements: 'which is two orders of magnitude lower than the 10° assumed in certain geodynamic researches.' (please see lines 14 and 15) and 'Despite the absence of confirmed observations of the ICW^{12,13} and the lack of universal acceptance of the excessive static tilt of 10° , the

possibility of a static-tilted inner core remains, and further investigation has been conducted to explore the impact of a static tilt on the period of the ICW¹².’ (please see lines 44-47).

Reference:

- Dehant, V., Hinderer, J., Legros, H., & Lefftz, M. (1993). Analytical approach to the computation of the Earth, the outer core and the inner core rotational motions. *Physics of the Earth and Planetary Interiors*, 76(3-4), 259-282.
- Ding, H., Pan, Y., Xu, X. Y., Shen, W., & Li, M. (2019). Application of the AR-z spectrum to polar motion: A possible first detection of the inner core wobble and its implications for the density of Earth's core. *Geophysical Research Letters*, 46(23), 13765-13774.
- Dumberry, M., & Bloxham, J. (2002). Inner core tilt and polar motion. *Geophysical Journal International*, 151(2), 377-392.
- Dumberry, M. (2008). Decadal variations in gravity caused by a tilt of the inner core. *Geophysical Journal International*, 172(3), 921-933.
- Dumberry, M. (2009). Influence of elastic deformations on the inner core wobble. *Geophysical Journal International*, 178(1), 57-64.
- Dumberry, M., & Manda, M. (2022). Gravity variations and ground deformations resulting from core dynamics. *Surveys in Geophysics*, 1-35.
- Greiner-Mai, H., Jochmann, H., & Barthelmes, F. (2000). Influence of possible inner-core motions on the polar motion and the gravity field. *Physics of the Earth and Planetary Interiors*, 117(1-4), 81-93.
- Greiner-Mai, H., & Barthelmes, F. (2001). Relative wobble of the Earth's inner core derived from polar motion and associated gravity variations. *Geophysical Journal International*, 144(1), 27-36.
- Guo, J. Y., Greiner-Mai, H., & Ballani, L. (2005). A spectral search for the inner core wobble in Earth's polar motion. *Journal of Geophysical Research: Solid Earth*, 110(B10).
- Mathews, P. M., Buffett, B. A., Herring, T. A., & Shapiro, I. I. (1991). Forced nutations of the Earth: Influence of inner core dynamics: 1. Theory. *Journal of Geophysical Research: Solid Earth*, 96(B5), 8219-8242.
- Mathews, P. M., Herring, T. A., & Buffett, B. A. (2002). Modeling of nutation and precession: New nutation series for nonrigid Earth and insights into the Earth's interior. *Journal of Geophysical*

Research: Solid Earth, 107(B4), ETG-3.

Rochester, M. G., Crossley, D. J., & Zhang, Y. L. (2014). A new description of Earth's wobble modes using Clairaut coordinates: 1. Theory. *Geophysical Journal International, 198(3)*, 1848-1877.

Moderate points

? L25-26, L31-32, L121-123 and figure 4: From the outset I think that the potential offset between the inner core rotation axis and the mantle rotation axis should be mentioned. L25-26 imply that they should be the same, but the main point of the paper is to argue that they are not. Therefore, the first mention of the inner core and mantle rotation axes should make that distinction and use the Ω_m and Ω'_m notation used elsewhere in the manuscript.

Response: Thanks for your reminder! We have modified the relevant parts as you suggested, please see lines 26, 27, 30, 31, 36, 38, and 41.

? I had not appreciated that the amplitude information from the AR-z process was not particularly meaningful, thank you for that clarification. I think that the background noise level indication in the figure included in the response should also be used in the paper (figure 2a) to illustrate whether the various peaks rise above the detection threshold. I remain concerned that the other peaks detected in the PM and LOD time series have no apparent physical explanation, but the reasoning for choosing that one peak as indicative of the ICW is now clearer.

Response: Thank you, it's a good idea to add the background noise level in Figure. 2, we have made these changes (please see line 121). As for the other signals, the ~5.9yr signal has been suggested as the inner core oscillation coupled with torsional wave in the Earth's core but still remains controversial (Buffett, 1996; Buffett and Mound, 2005; Gillet et al., 2010); Gillet et al. (2022) identified and modeled a non-axisymmetric ~7.3yr Magneto-Coriolis wave in the Earth's outer core that can explain the ~7.3yr signal in the Earth's rotation; the peak in the ~9-11yr is possible from the ~11yr Schwabe solar cycle; the period of ~13yr was also found in the geomagnetic dipole field (Jin and Thomas, 1977), but relevant physical mechanism has been not proposed; the ~18-23yr spectral peak may be mainly caused by the 18.6yr tidal signal and the ~22yr Hale solar cycle or high-latitude MAC (magnetic-Archimedes-Coriolis forces) wave (Nicolas and Buffett, 2023) in the Earth's core. Thus, we have made corresponding changes and added relevant physical explanations as your suggestions. Please see lines

99-106.

Reference:

- Buffett, B. A. (1996). Gravitational oscillations in the length of day. *Geophysical Research Letters*, 23(17), 2279-2282.
- Buffett, B. A., & Mound, J. E. (2005). A Green's function for the excitation of torsional oscillations in the Earth's core. *Journal of Geophysical Research: Solid Earth*, 110(B8).
- Gillet, N., Jault, D., Canet, E., & Fournier, A. (2010). Fast torsional waves and strong magnetic field within the Earth's core. *Nature*, 465(7294), 74-77.
- Gillet, N., Gerick, F., Jault, D., Schwaiger, T., Aubert, J., & Istaş, M. (2022). Satellite magnetic data reveal interannual waves in Earth's core. *Proceedings of the National Academy of Sciences*, 119(13), e2115258119.
- Jin, R. S., & Thomas, D. M. (1977). Spectral line similarity in the geomagnetic dipole field variations and length of day fluctuations. *Journal of Geophysical Research*, 82(5), 828-834.
- Nicolas, Q., & Buffett, B. (2023). Excitation of high-latitude MAC waves in Earth's core. *Geophysical Journal International*, 233(3), 1961-1973.

Minor points

? L14: "than that"  "than the"

Response: Done as you suggested. Please see line 14.

? L41: a citation to Triana et al (2022) might be included as another recent review relevant to the question of the importance of the topic.

Response: Thanks for your suggestion and sharing this paper with us, it is an insightful review paper worth reading. We have added this reference as your suggestion. Please see line 53.

? L89-90: "Since there is no other mechanism has been proposed..."  "Since no other mechanism has been proposed..."

Response: Done as you suggested. Please see line 108.

? L95: “super frequency resolution”, the use of “super” here feels awkward to me.

Response: Our apologies for using this inappropriate word, we have modified it as “high-frequency resolution”, please see line 113.

? L117: “ICWs”  “ICW signals” or similar

Response: Done as you suggested. Please see line 135.

? L167: “inner-mantle” is missing “core”

Response: Our apologies for this oversight. Given that the word “the mantle-inner core gravitational” appears many times, we chose to abbreviate it as “MICG” the first time it appears (please see line 178), so we use the abbreviation directly here (please see line 187).

References

- Buffett, B. A. Gravitational oscillations in the length of day. *Geophys Res Lett* 23, 2279–2282 (1996).
- Dumberry, M. & Manda, M. Gravity Variations and Ground Deformations Resulting from Core Dynamics. *Surv Geophys* 1–35 (2021) doi:10.1007/s10712-021-09656-2.
- Greiner-mai, H. & Barthelmes, F. Relative wobble of the Earth’s inner core derived from polar motion and associated gravity variations. *Geophys J Int* 144, 27–36 (2001).
- Triana, S. A. et al. Core Eigenmodes and their Impact on the Earth’s Rotation. *Surv Geophys* 43, 107–148 (2022).

REVIEWERS' COMMENTS

Reviewer #2 (Remarks to the Author):

The authors have answered properly my concerns and met my suggestions. I've also seen the changes made as a result of the other reviews. In general, I am satisfied with the revision and think that the observation of the ICW free mode of the earth rotation is very relevant and that the current version is suitable for publication.

I have still a few minor suggestions and I'd like to be sure that I've understood some points properly. They are:

1. The paper says nothing about the observed ICW oscillation being steady and showing no decay. This must be so because the ICW frequency must be real according to theory, but I'm afraid the contact of most people with free modes limits to the two ones known so far, CW and FCN, and both experience dissipation. I also think the property I said is not quite popular, since I have seen it published only in the proceedings of an IAU Colloquium. The series of papers coauthored by Mathews in 1991 and 2002 do not contain any theoretical approach to the topic of ICW dissipation nor give any estimation of its Q factor (they never even suggest a 0 value). I think it is a further evidence supporting you found the ICW signal, please consider it. The paper I referred to is coauthored by me, and I think it's not fair to point at my own papers in a review. Please don't feel you have to comment this point at all. It can be accessed at

A Escapa, J Getino, JM Ferrandiz (2000) Free frequencies for a three layered Earth model International Astronomical Union Colloquium 178, 481-485. https://www.cambridge.org/core/services/aop-cambridge-core/content/view/7BA4FC87472F405EDFOC8A848A1E9AE7/S0252921100061637a.pdf/free_frequencies_for_a_three_layered_earth_model.pdf

2. I'm not sure if I understand properly the meaning of figure 4b. I wonder if it is intended to display the location of the torque in a generic case, of a special position of it, namely an extreme. My reasoning for asking this is the following: the torque equatorial projection must rotate keeping perpendicular to the equatorial angular velocity that describes a circumference under the ICW oscillation, so that it cannot be fixed in the equator. However, if the SIC axis lies in the meridian plane containing 90W, 90E, when PM_ICW is in the perpendicular (containing the 0 meridian) y -PM vanishes and $dLOD/dt$ also because you report its observed variation at the ICW period is in phase with y -PM. Therefore, when PM_ICW is in the 90W-90E plane $dLOD/dt$ will be maximum/minimum and the z-torque too because of the $\pi/2$ phase shift. In

such a case, the bars applied to the torques should mean “extreme”, but I haven’t seen it in the text. Please, explain me if this is true or I missed or misunderstood something.

What I have difficulties to fully understand is why the 3D-torque must lie in the SIC equator and not in another plane passing by the same line of nodes, please clarify it further to me.

I take this opportunity to say that I liked the related explanation of the R1 version (lines 159-166) better than the new R2 one (177-186).

Reviewer #3 (Remarks to the Author):

Thank you for your detailed and lengthy response, and for making the changes to the manuscript. Your reasoning and notation are now much clearer. The determination of the IC static tilt and ICB density jump will be of interest to the deep Earth research community.

I have only a few minor comments on the latest version.

line 21: "flat solid Earth", the word flat here is confusing. Do you mean "laterally homogeneous"?

line 43: "is proposed to coincide" -> "was proposed to coincide" as this is reporting on a previous study.

Figure 2: Thank you for adding the grey bar to the figure to indicate the level required to rise above background noise. I think that the meaning of this region should be included in the caption.

Line 177: "forms a tilted" might be better as "results in a tilted"

Line 178: "which has" might be better as "having"

Line 181: "always in the plane" -> "is always in the plane"

Line 177-183: This is a single lengthy sentence. It might read better if it was broken into two or three.

Responses to Reviewers

#####

Reviewer #2 (Remarks to the Author):

The authors have answered properly my concerns and met my suggestions. I've also seen the changes made as a result of the other reviews. In general, I am satisfied with the revision and think that the observation of the ICW free mode of the earth rotation is very relevant and that the current version is suitable for publication.

I have still a few minor suggestions and I'd like to be sure that I've understood some points properly.

They are:

1. The paper says nothing about the observed ICW oscillation being steady and showing no decay. This must be so because the ICW frequency must be real according to theory, but I'm afraid the contact of most people with free modes limits to the two ones known so far, CW and FCN, and both experience dissipation. I also think the property I said is not quite popular, since I have seen it published only in the proceedings of an IAU Colloquium. The series of papers coauthored by Mathews in 1991 and 2002 do not contain any theoretical approach to the topic of ICW dissipation nor give any estimation of its Q factor (they never even suggest a 0 value). I think it is a further evidence supporting you found the ICW signal, please consider it. The paper I referred to is coauthored by me, and I think it's not fair to point at my own papers in a review. Please don't feel you have to comment this point at all. It can be accessed at A Escapa, J Getino, JM Ferrandiz (2000) Free frequencies for a three layered Earth model International Astronomical Union Colloquium 178, 481-485. https://www.cambridge.org/core/services/aop-cambridge-core/content/view/7BA4FC87472F405EDF0C8A848A1E9AE7/S0252921100061637a.pdf/free_frequencies_for_a_three_layered_earth_model.pdf

Response: Thanks for your support for our study and your kind reminder about the Q value of ICW. Although you think that we do not have to reply, we are very happy to see your reminder. As you mentioned, there is currently little research about the Q value of ICW, and the calculation of (theoretical and observational) Q value of ICW is another important work and needs further study. The literature you provided is of great reference significance, and we will refer to it when we do relevant research about the Q of ICW in the future.

2. I'm not sure if I understand properly the meaning of figure 4b. I wonder if it is intended to display the location of the torque in a generic case, of a special position of it, namely an extreme. My reasoning for asking this is the following: the torque equatorial projection must rotate keeping perpendicular to the equatorial angular velocity that describes a circumference under the ICW oscillation, so that it cannot be fixed in the equator. However, if the SIC axis lies in the meridian plane containing 90W, 90E, when PM_ICW is in the perpendicular (containing the 0 meridian) y-PM vanishes and dLOD/dt also because you report its observed variation at the ICW period is in phase with y-PM. Therefore, when PM_ICW is in the 90W-90E plane dLOD/dt will be maximum/minimum and the z-torque too because of the $\pi/2$ phase shift. In such a case, the bars applied to the torques should mean "extreme", but I haven't seen it in the text. Please, explain me if this is true or I missed or misunderstood something.

What I have difficulties to fully understand is why the 3D-torque must lie in the SIC equator and not in another plane passing by the same line of nodes, please clarify it further to me.

I take this opportunity to say that I liked the related explanation of the R1 version (lines 159-166) better than the new R2 one (177-186).

Jose M Ferrandiz

Response: Thanks for your thoughtful review and insightful comments. You have got the message properly. Here, we will further explain that the 3D-torque must lie in the SIC equator and not in another plane passing by the same line of nodes. Grey shaded plane represents the mantle's equatorial plane, and it is perpendicular to the figure (or rotation) axis of the mantle. Blue shaded plane represents the inner core's equatorial plane, and it is perpendicular to the rotation axis (or mean figure axis) of the inner core. The mean figure axis of the inner core should be aligned with the internal axis of the mantle (or the mean figure axis of the lower mantle). This is because the inner core and lower mantle are closer and therefore they have a stronger gravitational pull. Because of the density imbalance or asymmetry in the Earth's interior, the mantle results in a tilted internal axis relative to its rotation axis, and the mantle-inner core system will have the lowest gravitational potential energy when the mean figure axis of the inner core is aligned with the tilted internal axis of the mantle. When the inner core wobbles and deviates from this tilted axis, the MICG torque forces the inner core back to this tilted axis. Thus, MICG torque is in a plane perpendicular to this tilted axis, i.e., MICG torque is in the blue-shaded plane rather than the grey shaded plane. The internal axis of the heterogeneous mantle is

innovative, and we have developed a detailed mathematical formulation after discussions with Benjamin F. Chao. Considering your comments and the suggestions of Reviewer# 3, we have made simple changes, please see lines 158-165.

#####

Reviewer #3 (Remarks to the Author):

Thank you for your detailed and lengthy response, and for making the changes to the manuscript. Your reasoning and notation are now much clearer. The determination of the IC static tilt and ICB density jump will be of interest to the deep Earth research community.

Response: Thank you very much for your recognition of our work and positive assessment. We have carefully considered your suggestion and revised this MS according to the comments, and hope that the revised version is more suitable.

I have only a few minor comments on the latest version.

line 21: "flat solid Earth", the word flat here is confusing. Do you mean "laterally homogeneous"?

Response: Thank you very much for your kind reminder, we think that "oblate" is a more appropriate expression, please see line 21.

line 43: "is proposed to coincide" -> "was proposed to coincide" as this is reporting on a previous study.

Response: Thank you for your kind reminder, we have changed it in the new MS, please see line 43.

Figure 2: Thank you for adding the grey bar to the figure to indicate the level required to rise above background noise. I think that the meaning of this region should be included in the caption.

Response: Thank you for your kind reminder, we have added a note about this according to your suggestions, please see line 464.

Line 177: "forms a tilted" might be better as "results in a tilted"

Response: Done as you suggested. Please see line 158.

Line 178: "which has" might be better as "having"

Response: Thanks for your suggestions, we have changed this long sentence into several clauses according to your last suggestion. Please see line **159**.

Line 181: "always in the plane" -> "is always in the plane"

Response: Done as you suggested. Please see line **163**.

Line 177-183: This is a single lengthy sentence. It might read better if it was broken into two or three.

Response: Thank you for the suggestions to make it more readable, we changed it as you suggested, please see lines **158-165**.